# On the distinctiveness of observed oceanic raindrop distributions

David Ian Duncan[1], Patrick Eriksson[1], Simon Pfreundschuh[1], Christian Klepp[2], and Daniel C. Jones[3]

[1]Department of Earth, Space, and Environment, Chalmers University of Technology, SE 412 96 Gothenburg, Sweden
[2]CliSAP/CEN, Meteorological Institute, Universität Hamburg, 20146 Hamburg, Germany.
[3]British Antarctic Survey, CB3 0ET Cambridge, United Kingdom

**Correspondence:** David Ian Duncan (david.duncan@chalmers.se)

**Abstract.** Representation of the drop size distribution (DSD) of rainfall is a key element of characterizing precipitation in models and observations, with a functional form necessary to calculate the precipitation flux and the drops' interaction with radiation. With newly available oceanic disdrometer measurements, this study investigates the validity of commonly used DSDs, potentially useful a priori constraints for retrievals, and the impacts of DSD variability on radiative transfer. These
data are also compared with leading satellite-based estimates over ocean, with the disdrometers observing more small drops and significantly more variability in number concentrations. This indicates that previous appraisals of raindrop variability over ocean may have been underestimates. Forward model errors due to DSD variability are shown to be significant for both active and passive sensors. The modified gamma distribution is found to be generally adequate to describe rain DSDs, but may cause systematic errors for high latitude or stratocumulus rain retrievals; depending on the application, an exponential or generalized
gamma function may be preferable for representing oceanic DSDs. An unsupervised classification algorithm finds a variety of DSD shapes that differ from commonly used DSDs, but does not find a singular set that best describes the global variability.

## 1 Introduction

A challenge shared by atmospheric models and remote sensing retrievals alike is the representation of precipitation microphysics. Raindrops can be modeled using a variety of functional forms, simple relations between drop size and number density
that attempt to capture the overall behavior in a way sufficient to represent the processes of interest. The radiative characteristics and precipitation flux through an atmospheric volume containing precipitation depend on the size and resulting terminal velocities of the rain drops, defined via that volume's drop size distribution (DSD). In this way, the DSD acts as a necessary conduit to represent precipitation processes, one common to climate models, radar retrievals, and data assimilation schemes.

Various functional forms have been employed to describe rain DSDs. Exponential DSDs (Marshall and Palmer, 1948) have
been used in radar meteorology for decades, and different versions of the modified gamma distribution (MGD; Eq. 1) have gained popularity for remote sensing (Ulbrich, 1983). Simplifications of the MGD to three, two, or single parameter versions yield the gamma, exponential, and power law relations (Petty and Huang, 2011), respectively, all of which are used to represent DSDs in various applications. Note that the four parameter MGD is sometimes called the generalized gamma distribution (Petty and Huang, 2011; Thurai and Bringi, 2018). Between those who use the MGD to describe DSDs, disagreement exists on how
many free parameters to use (Smith, 2003; Thurai and Bringi, 2018), whether it is best to normalize the distribution (as in

Eq. 3) in some way (Testud et al., 2001), or if the separation of parameters in the MGD is either physically meaningful or outperformed by simpler methods (Williams et al., 2014; Tapiador et al., 2014).

The below equations will be referred to throughout the text as the generic MGD function (Eq. 1) and normalized gamma (NG) function (Eq. 3), with NG a normalized and 3-parameter version of the MGD. Here $N(D)$ is the number of drops per
volume per size as a function of the drop diameter, $D$ (with $D$ given in $\mathrm{mm}$ and $N(D)$ in $\mathrm{mm^{-1}\,m^{-3}}$). The so-called shape parameter is $\mu$ while $N_0$ and $N_w$ are intercept parameters, with $N_w$ (Eq. 5) a normalized intercept parameter scaled by the water content (Testud et al., 2001). The mass-weighted mean diameter is $D_m$ (Eq. 2), the ratio of the fourth and third moments of the distribution (Eq. 2). Denoting the gamma function is $\Gamma$, $\rho_w$ is the density of water, and $RWC$ is the rain water content in $\mathrm{kg\,m^{-3}}$.

$$N(D) = N_0 D^\mu e^{-\lambda D^\gamma} \tag{1}$$

$$D_m = \frac{\int_0^\infty N(D)D^4 dD}{\int_0^\infty N(D)D^3 dD} \tag{2}$$

$$N(D) = N_w f(\mu)(\frac{D}{D_m})^\mu e^{-(4+\mu)\frac{D}{D_m}} \tag{3}$$

$$f(\mu) = \frac{\Gamma(4)}{4^4} \frac{(4+\mu)^{4+\mu}}{\Gamma(4+\mu)} \tag{4}$$

$$N_w = \frac{4^4}{\pi \rho_w} \frac{RWC}{D_m^4} \tag{5}$$

Scattering of radiation is highly dependent on particle size, and thus the DSD is a crucial component of remote sensing retrievals, whether it is assumed or retrieved. Depending on the application, the specific choice of DSD may or may not make much difference (Smith, 2003; Illingworth and Blackman, 2002). For instance, erroneous assumptions about small drops may
not impact the broadband radiative fluxes or precipitation characteristics of a volume, but a more accurate DSD representation may be necessary when considering additional frequencies or polarized measurements. The under-constrained nature of precipitation retrieval means that the DSD is either assumed completely or needs to be constrained to allow tractable solutions.

A lack of global DSD data has hampered the retrieval of precipitation from satellites. Satellite retrievals rely heavily on a priori knowledge to constrain the solution space, and regional differences in meteorology and microphysics can manifest
as regional biases in satellite retrievals (Berg et al., 2006). Whereas ground radar networks and arrays of disdrometers over land have helped to characterize the variability of raindrops from continental precipitation (Bringi et al., 2003; Williams and Gage, 2009; Thurai and Bringi, 2018), observations of DSDs over ocean have mostly been limited to field campaigns, a few

small tropical islands and atolls, and coastal radar retrievals. Because of the different aerosol loading, convective strength, and underlying humidity of airmasses over land, oceanic drop populations can be distinct from those over land (Dolan et al., 2018), with the different microphysics influencing satellite retrievals. It is thus desirable to have measurements of DSDs over ocean, and crucial that these measurements are globally representative rather than skewed toward one region or another.

It is expedient to condense the variability of DSDs into a few distinct classes, either to narrow the possible solution space of remote sensing retrievals or for interpretation of results. Separation of stratiform and convective precipitation has long been common, as stratiform precipitation tends to have a more peaked distribution of fewer, smaller drops versus the more exponential distribution of precipitation from convective clouds (Thurai et al., 2010; Thompson et al., 2015). However, partitioning stratiform and convective rainfall is done in various ways and may differ depending on location. A little further, Dolan et al.
(2018) argue for six dominant modes of DSDs globally, separated via principal component analysis but linked to meteorology and attendant microphysical regimes. As many studies of drop distributions are from land-based disdrometers and radars, DSD variability has been studied less over open ocean where a majority of global precipitation occurs, though advances are being made in this area (Thompson et al., 2018).

    In remote sensing applications, one can attempt to solve for all, some, or none of the parameters that define a functional
form such as Eq. 1, depending on the information content available. A normalized distribution such as Eq. 3 is used in many precipitation retrievals to separate the water content from the spectrum's shape. In that formulation with RWC separate, this leaves two free parameters to define the distribution since RWC is directly related to $N_w$ through $D_m$. While passive-only retrievals may need to assume one of these parameters because of the limited signal available (Duncan et al., 2018), radar or combined radar/radiometer retrievals may solve for these parameters in a constrained way (Munchak et al., 2012; Grecu et al.,
2016). Precipitation retrievals thus handle the complexity of the DSD differently depending on their instruments' sensitivities, but necessarily using a predefined functional form to limit the inverse problem's degrees of freedom.

    To investigate the distinctiveness of raindrop size distributions over the global oceans, and how this may impact retrievals both in terms of prior constraints and radiative transfer modeling, the study proceeds as follows. Data and methods are described in the next section, introducing the disdrometer and satellite data examined, as well as the machine learning technique
used to classify drop regimes. Section 3 presents a holistic view of global disdrometer measurements with respect to the normalized gamma distribution, including a comparison to the leading satellite-based, near-global DSD data set. Results from the application of a machine learning technique to the disdrometer data are discussed in Section 4. In Section 5 the radiative aspects of DSD variability are addressed in the context of satellite retrievals with radiative transfer modeling. Then follows a discussion section, critically examining the disdrometer data versus a commonly used functional form. The paper closes with
a summary and some conclusions.

**Table 1.** Center drop diameters for OceanRAIN size bins, given in mm. Note that the bins of up to 5 mm diameter are given here for brevity and because these bins contain the vast majority of drop counts, but larger size bins also exist. The values are rounded to 3 digits. For full details see Klepp et al. (2017).

| | | | | | | | | | |
|---|---|---|---|---|---|---|---|---|---|
| .392 | .427 | .462 | .498 | .535 | .573 | .612 | .652 | .693 | .735 |
| .778 | .823 | .868 | .914 | .961 | 1.01 | 1.06 | 1.11 | 1.16 | 1.22 |
| 1.27 | 1.33 | 1.39 | 1.45 | 1.51 | 1.57 | 1.63 | 1.70 | 1.76 | 1.83 |
| 1.90 | 1.97 | 2.05 | 2.12 | 2.20 | 2.28 | 2.36 | 2.45 | 2.53 | 2.62 |
| 2.71 | 2.80 | 2.89 | 2.99 | 3.09 | 3.19 | 3.30 | 3.40 | 3.51 | 3.62 |
| 3.74 | 3.86 | 3.98 | 4.10 | 4.23 | 4.36 | 4.49 | 4.62 | 4.76 | 4.91 |

## 2 Data and Methods

### 2.1 OceanRAIN

The Ocean Rainfall And Ice-phase precipitation measurement Network (OceanRAIN) coordinates disdrometer measurements and acquired ancillary data aboard research ships across the global oceans (Klepp et al., 2018). The data set begins in 2010 and collection is ongoing, with observations spanning 8 vessels and over 6 million minutes covering all ocean latitudes. Ocean-RAIN data contain raw counts integrated for each minute of rain, snow, or mixed-phase precipitation, with derived rainfall DSD parameters (Eq. 3), and various ancillary fields. The large and growing size of the data set make statistical analysis possible due to its consistent application across various ships. The disdrometer data are integrated per minute and separated into logarithmically-spaced size bins (Table 1), permitting analysis of DSDs without the assumption of a functional form. Specifically, the OceanRAIN-M ("OceanRAIN Microphysics") data are used primarily in the study (Klepp et al., 2017), in which drop counts from the disdrometer are converted to number concentrations per size (i.e. drops per volume per size), the form in which DSDs are often given. DSD assumptions commonly made in the literature can thus be assessed against the observations.

Underpinning OceanRAIN is the ODM470 optical disdrometer, a sensor with sensitivity to hydrometeors of diameter 0.4 to 22 mm (Klepp, 2015). The disdrometer is deployed on the superstructure of ships in a package including a cup anemometer and a precipitation detector to activate the disdrometer. A wind vane turns the disdrometer to keep the optical path normal to the wind direction, and the disdrometer's cylindrical volume ensures that the incident angle of hydrometeors does not affect the measurement. These work in concert to minimize impacts of turbulence from local up- and down-drafts, to limit under-catchment and drops impacting the sensor from various directions (Klepp, 2015). Only data points marked as rain definite, with 50 or more measured drops, and with a probability of precipitation of 100% were used in the following analysis. To be consistent, only data points with measurements in ten or more size bins are used (Klepp et al., 2018), as these provide the parameters from the NG fit to Eq. 3. A visualization of the OceanRAIN sampling used in this study is found in Fig. 1, with raining minutes shown on a near-global regular grid.

This study also makes use of simulated reflectivites in Section 5. Simulated reflectivites from the ODM470 disdrometer have demonstrated high correlation and a near-zero bias when compared against co-located, vertically-oriented radar observations (Klepp et al., 2018, Fig. 6). In comparisons with co-located rain gauges, the optical disdrometer performs better in high wind speeds, as under-catch is a significant problem for traditional rain gauges that can result in underestimation of rainfall accumulation by 50% (Grossklaus et al., 1998; Klepp et al., 2018), though accumulations match within 2% for low wind speeds (Klepp, 2015). The ODM470 has been used in a variety of conditions and shown no difference in accuracy between oceanic and continental cases (Bumke and Seltmann, 2011).

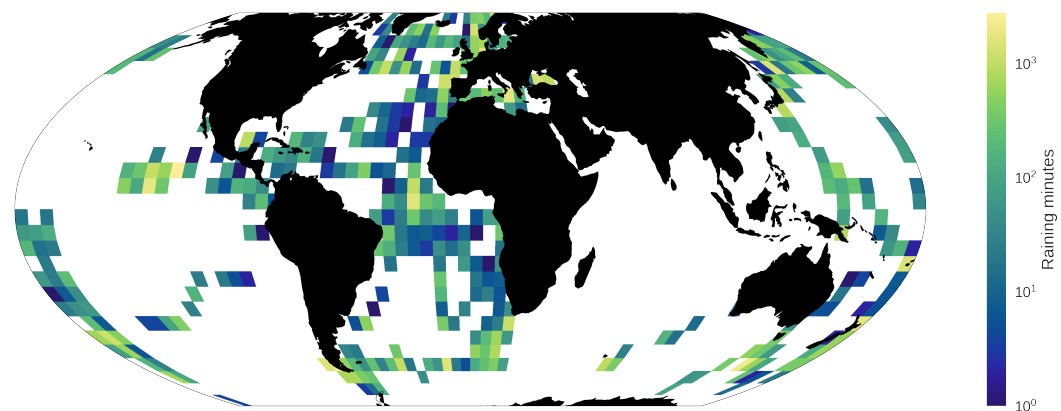

**Figure 1.** Raining minutes from OceanRAIN, selected by the sampling criteria of this study as described in Section 2.2, viewed on a regular $5°$ grid. Grid cells in white signify that no data points were used.

The accuracy of disdrometer-derived DSD parameters (following Eq. 3) will depend somewhat on the parameter discussed and the type of rain. For instance, derived $D_m$ should be accurate for all but the weakest rain rates as it is simply defined (Eq. 2) and requires no fitting. The accuracy of derived $N_w$ may be suspect for cases with high rain rates and a low $D_m$ value, as drops below the sensitivity threshold may constitute a non-negligible fraction of total drops, though this depends on the type of rainfall and is an issue faced by all disdrometers (Thurai et al., 2017). To be clear, there can be significant number concentrations below this sensitivity limit, but voltages corresponding to drop diameters of less than $0.36\,\mathrm{mm}$ are disregarded as these can be contaminated by vibrations from the ship (Klepp et al., 2018) and this is a key drawback of the data set. The derived shape parameter, $\mu$, is the least reliable of the three as it depends on a curve fitting which may not be optimal for light rain rates or spectra that do not conform to the expected general shape. In other words, the accuracy of DSD parameters reported by OceanRAIN may exhibit bias in regimes with many drops below the disdrometer's sensitivity threshold, or for distributions with a shape unlike that assumed.

In this study, the default way of discussing the OceanRAIN data is using the 3-parameter normalized gamma distribution (Eq. 3), but a strength of this data set is that number concentrations are provided for every observed size bin (Table 1), allowing investigation of different DSD types, including other varieties of the MGD. Later in the study the 3-parameter MGD (i.e. NG, as all DSDs discussed are normalized) is contrasted with 1- and 2-parameter MGD versions as well as DSDs not conforming to

the MGD but instead derived from a machine learning technique. In the context used here, the 1-parameter MGD is equivalent to single moment microphysics in model parlance, with a fixed shape (i.e. prescribed $D_m$ and $\mu$) and $N_w$ simply scaling with $RWC$. The 2-parameter MGD is defined by a calculated $D_m$ (via Eq. 2) but a prescribed $\mu$, whereas the 3-parameter MGD includes calculated $D_m$ and $\mu$, with $N_w$ determined via $D_m$ and $RWC$ (Eq. 5).

## 2.2  GPM Combined Radar-Radiometer Algorithm

The Global Precipitation Measurement (GPM) Core Observatory (Hou et al., 2014) holds two sensors designed to measure precipitation: the GPM Microwave Imager (GMI) and the Dual-frequency Precipitation Radar (DPR). GMI is a passive microwave radiometer measuring from 10 to 190 GHz and the DPR is a phased array radar measuring at $K_U$ and $K_A$ bands (13.6 and 35.5 GHz, respectively). The dual frequencies of DPR set it apart from other satellite-borne sensors as far as the capacity to solve for the DSD. The GPM core satellite's combination of passive and active sensors provides sensitivity to a large range of precipitating hydrometeors, with information on their emission and scattering characteristics. However, DPR has limited sensitivity to small drops and low number concentrations due to its minimum detectable signal of 12 to 13dBZ.

The GPM Combined Radar-Radiometer Algorithm (Grecu et al., 2016), hereafter referred to as GPM CORRA, is a retrieval that uses data from both radar and radiometer to solve for profiles of hydrometeors that optimally fit the observations. As the GPM satellite represents the best observational platform yet flown for measuring near-global precipitation, the combined retrieval from DPR and GMI is included in this study as the state of the art for calculating global DSD statistics. Via the same DSD formulation given in Eq. 3, GPM CORRA first uses the $K_U$ band reflectivities to solve for the $D_m$ profile. It then retrieves $N_w$ at a reduced vertical resolution to match the $K_A$ band reflectivities, DPR path integrated attenuation, and de-convolved GMI brightness temperatures ($T_B$s) using optimal estimation. The shape parameter is fixed at $\mu = 2$ for all cases. For further details about this retrieval, see Grecu et al. (2016).

In this study, gridded level 3 GPM CORRA data are used (Olson, 2017), comprising monthly and daily files from GPM version V06. This data set provides statistics of pixel-level derived DSD parameters from Eq. 3 at $5°$ and $0.25°$ horizontal resolution. The values used in this study are from the lowest altitude bin and include oceanic pixels only so as to best match the surface-based data from OceanRAIN-M. Because GPM CORRA receives most of its information content from DPR, the DSD parameters derived are representative of individual segments of the atmospheric column and not a column average, a key difference from passive-only retrievals. This is significant, as comparison with surface-based observations (Petersen et al., in press) should be as close in altitude as possible, as DSDs will vary with altitude as evaporation, coalescence, collisions, or other processes modify the spectra (Williams, 2016). The $250\,\mathrm{m}$ vertical resolution of DPR means that multiple observations exist below $1\,\mathrm{km}$ altitude, though some of these will be affected by surface clutter and so the lowest bin without clutter is chosen here. Note that the GPM CORRA retrievals were performed at the native DPR pixel size, which has a $5\,\mathrm{km}$ horizontal resolution.

To assess the similarity between GPM estimates and the in situ disdrometer measurements of OceanRAIN, in Section 3.2 the retrieved results for $N_w$ and $D_m$ are compared, as GPM CORRA assumes a constant $\mu$ value. To perform this comparison, level 3 GPM CORRA data were used, spanning 12 months from 2017. Due to the uneven sampling of the ship-borne disdrometers,

GPM data included in the analysis are from months with valid OceanRAIN data points in each box and defined as ocean pixels by DPR. No attempt was made to match observations exactly in space and time due to the difficulty of point-to-area comparisons with ship-borne data and GPM (Burdanowitz et al., 2018; Loew et al., 2017).

## 2.3 Gaussian Mixture Modeling

Gaussian Mixture Modeling (GMM) is an unsupervised, probabilistic classification technique that attempts to represent a data set using a linear combination of multidimensional Gaussians in a chosen parameter space (Pedregosa et al., 2011). The dimensions (or "features") of the parameter space and the maximum number of classes, $N_{GMM}$, are set by the user. GMM assigns each data point to the class, represented by a multidimensional Gaussian function, with the highest posterior probability for that data point. For further technical details on GMM and its use in other Earth science applications, see Maze et al. (2017)
and Jones et al. (2019).

GMM generalizes to a wide variety of data distributions and can thus identify structures in the DSD data that might be missed by more traditional classification methods. This frees the analysis from explicit assumption of a DSD shape such as Eq. 3. In the approach used here, the dimensions given to the GMM module are the size bins used by the OceanRAIN disdrometers and thus the input data are an array of approximately 90000 raining minutes with 60 size bins. These data are unchanged other
than being normalized so that DSD "shape" variability in the data set is not weighted by the total number of drops observed, and cut off at 60 size bins as very few drops over $5\,\mathrm{mm}$ are ever measured. Because the shapes are independent of the total number of drops, this is analogous to the normalized DSD approach typified by Eq. 3. GMM thus finds common shapes of the observed DSDs and determines the posterior probability of every data point (DSD for each raining minute) falling into each of the various classes. Each observed DSD is assigned to the GMM class for which it has the highest posterior probability. The
resultant classes provide insight into dominant structures of the input data, with this approach exemplified in Section 4.

The number of GMM classes is set a priori, with the degree of complexity described by the GMM decomposition dependent on the number of states set by the user. Determining an optimal value for $N_{GMM}$ is thus important but somewhat subjective because the desired level of complexity retained after the decomposition will vary for different applications. One method for estimating a suitable range for the number of classes is to use the Bayesian Information Criterion (Schwarz, 1978). Shown in
Eq. 6, this metric (BIC) contrasts the log likelihood (L) against a cost for the number of classes (K) to provide an objective measure of how many classes should optimally describe the data, where $N_f(K) = K - 1 + KD + KD(D-1)/2$, with $D$ the dimension of the data space and $n$ the number of data points used in model training. The first term in Eq. 6 becomes more negative with increased likelihood, while the second term acts to penalize overfitting. The minimum BIC thus signifies the optimal K value, maximizing the variability explained with the fewest possible classes. A plateau of BIC values versus
K would signify no distinctly optimal K to describe the data's variability, but rather a range of solution spaces in which the addition of further states provides marginal additional complexity.

$$BIC(K) = -2L(K) + N_f(K)log(n) \qquad (6)$$

# 3 Global results

## 3.1 Disdrometer results

Viewing the OceanRAIN data all together can provide a sense of the variability in DSD populations over the world's oceans. From the perspective of global retrievals, constraints on the DSD that depend on the location or environmental regime, rather than, say, partitioning stratiform and convective precipitation a priori, are useful for independent satellite-based products that do not ingest detailed model data, such as the operational retrievals for the GPM constellation radiometers (Kummerow et al., 2015). To this end, the derived parameters of Eq. 3 are given for all raining disdrometer observations, separated by latitude and SST in Fig. 2 and representing all data points shown in Fig. 1. As this is the DSD form most used in rainfall retrievals currently, it is presented here.

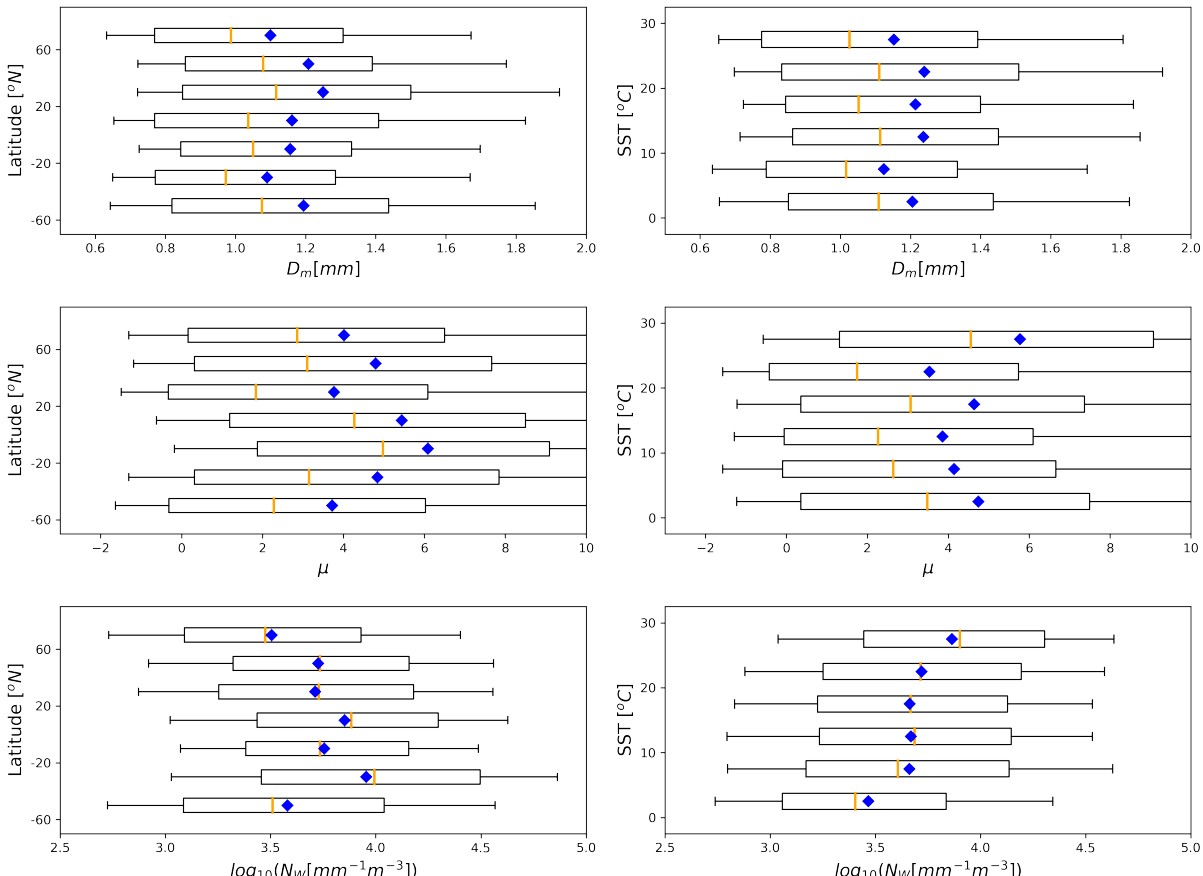

**Figure 2.** Distribution of DSD parameters following Eq. 3. The boxes define the standard deviations ($\pm 1\sigma$), the whiskers define the 10% and 90% bounds, orange lines denote the median, and blue diamonds the mean. Observations are divided according to latitude with $20°$ bins (left) and sea surface temperature with $5°$C bins (right).

As seen in Fig. 2, the normalized gamma DSD parameters exhibit a wide range of variability that is not strongly tied to latitude or SST. The strongest trend visible is that higher number concentrations occur over warmer ocean surfaces, with the mean $log_{10}(N_w)$ increasing from about 3.5 to 4.0, as may be expected due to the Clausius-Clapeyron equation. This is roughly in line with the a priori $N_w$ used for rain by Mason et al. (2017) of $3.9e3$, or 3.59 in log space. It is noted that the distributions of $D_m$ and $\mu$ are not Gaussian, with the means and medians separate, and $N_w$ only moderately Gaussian in log space.

It is stressed that OceanRAIN observations are not evenly distributed around the global oceans and thus the values seen are dependent on the sampling (i.e. where the ships sailed, see Fig. 1), so these values are not fully representative of each ocean latitude band. As surface-based observations, they do not provide information as to any vertical DSD variability, a topic that requires radar observations (Williams, 2016). However, it is possible to pick out some meteorological regimes of interest from the derived DSD parameters in OceanRAIN. For instance, the ships' heavy sampling of Southern Hemisphere stratocumulus regions (Fig. 1 shows up in these plots as a regime characterized by a higher number of small drops and a more peaked distribution (seen in the 20°S to 40°S band in Fig. 2). From the perspective of satellite rainfall retrievals, such location- or cloud regime-dependent a priori constraints are much preferable to a global prior and useable within existing algorithms.

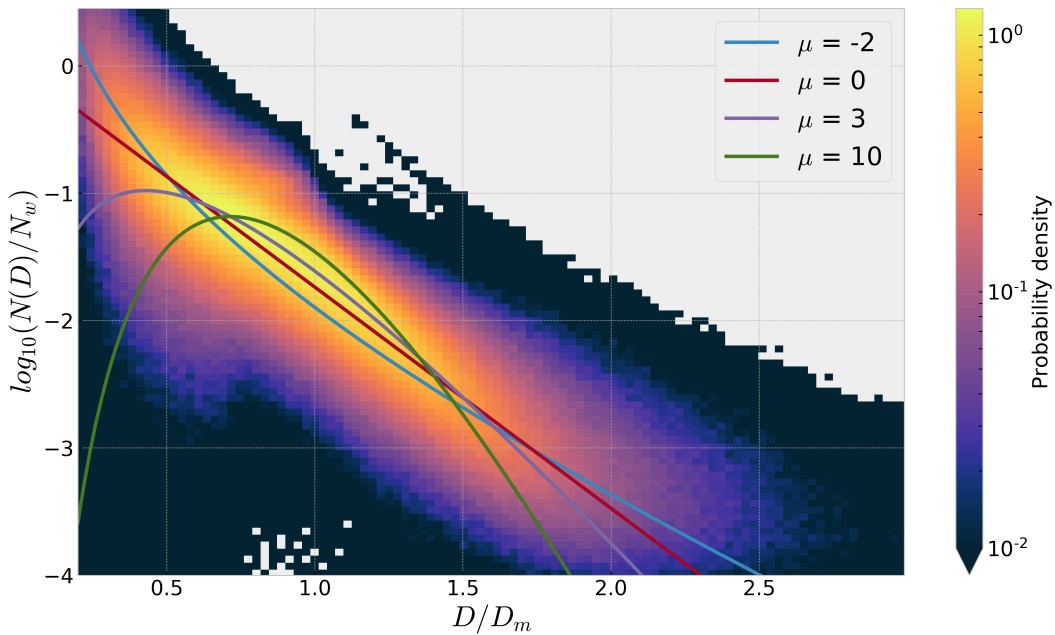

**Figure 3.** Probability density function of all raining OceanRAIN data points, visualized using the scaled DSD, $N(D)/N_W$, against the diameter normalized by $D_m$. Various curves with prescribed $\mu$ values are plotted for comparison. Areas in gray indicate no data.

Without applying any sorting methods or functional forms to the OceanRAIN data, it is worth viewing the data as a whole to see how closely the overall behavior resembles the MGD, as this is commonly used in the literature. Figure 3 shows a two-

dimensional probability density function (PDF) of drop diameter normalized by $D_m$ versus number concentration normalized by $N_w$. This is a view of drops' overall behavior often used to justify usage of the NG for precipitation (Bringi et al., 2003; Leinonen et al., 2012), as it permits visualization of in situ data points with the NG for various $\mu$ values including the exponential DSD. Figure 3 indicates that much of the spectral power within OceanRAIN lies near the exponential ($\mu$=0) line or near the lines with near-zero shape parameters. This is consistent with the enduring popularity of exponential DSDs and the $\mu = 2$ assumption of GPM CORRA.

## 3.2 Comparison to GPM CORRA

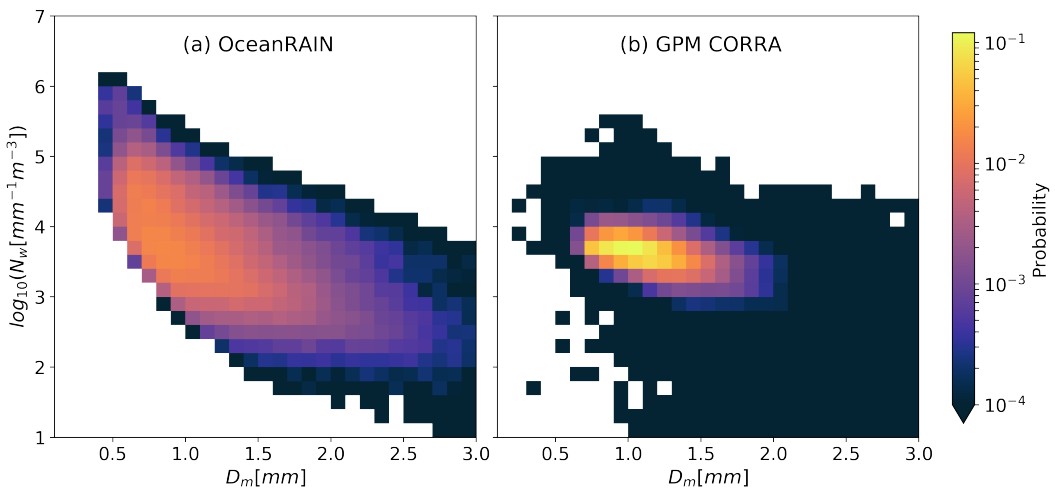

**Figure 4.** Probability histograms for OceanRAIN (a) and GPM CORRA (b) for observations of the normalized intercept parameter ($N_w$) and the mass-weighted mean diameter ($D_m$). Areas in white indicate no data.

Figure 4 shows two-dimensional histograms of $N_w$ versus $D_m$ for both OceanRAIN and GPM CORRA. Both datasets exhibit an inverse relationship between $N_w$ and increasing $D_m$ and show maximum probabilities of occurrence in the same area, namely near $D_m = 1mm$ and $log_{10}(N_w) = 3.8$. The disdrometers show greater spread in both parameters, but especially in $N_w$. Whereas both data sets observe most occurrences of $D_m$ between about 0.6 to $1.8\,\mathrm{mm}$, the range of $N_w$ observed by the disdrometers is easily twice as large, even in log space. This behavior is also seen in Fig. 5.

The left panel of Fig. 5 shows histograms of derived $D_m$ from the disdrometers compared with GPM CORRA, separated by latitude, with each latitude band 20 degrees wide. Given the limited sensitivity of DPR to small drops, it is unsurprising to note that OceanRAIN observes a wider distribution of $D_m$ that is clearly different from GPM results for small drops. Another key feature of these histograms is that while the maxima in $D_m$ distributions are relatively similar for the two data sets, OceanRAIN observes a less peaked distribution in most latitude bands. The disdrometers observe more small drops in all latitude bands, but this is especially pronounced in the Southern Ocean. For all latitudes GPM exhibits a peak near $D_m = 1\,\mathrm{mm}$ or just below.

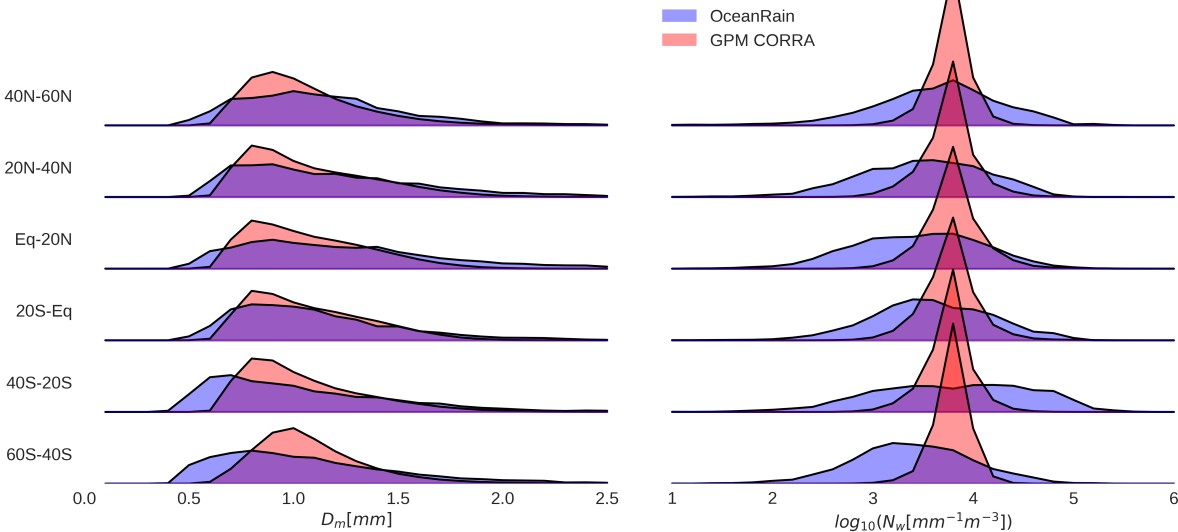

**Figure 5.** Normalized histograms of $D_m$ (left) and $log_{10}(N_w)$ (right) for GPM CORRA and OceanRAIN, separated by latitude. All histograms use a linear y-axis of height 20%. GPM data are from the 3B CMB monthly gridded product.

The right panel of Fig. 5 follows the same format but for derived $N_w$. The most striking aspect of these histograms is the strongly peaked distribution retrieved by GPM in all latitude bands. In contrast, the disdrometers observe many cases with $N_w$ values an order of magnitude greater or smaller than those of the GPM distributions. The peak $N_w$ values from the disdrometers are similar to those of GPM CORRA in the northernmost latitude band, but are significantly wider and flatter in every latitude
band shown.

## 4    GMM-derived states

As shown in Fig. 3, the NG with a low $\mu$ value lies near the highest probability densities of the observed PDF. However, a great deal of spread exists that is not captured by any one curve. With this in mind, GMM was employed to investigate if a finite number of DSD shapes without a predefined functional form could better capture this variability.

To provide a visualization of how the GMM states attempt to fit the observed DSD from the disdrometer, and how these states compare with various MGD forms, Fig. 6 contains randomly sampled data points from OceanRAIN. These four data points have quite different rain rates and RWCs. The GMM curves shown are from iterations with $N_{GMM}$ of 6 and 14, two of the panels given in the subsequent figure; these are the states with the highest posterior probability from GMM, indicating the best match to the observed distribution. No fitting was performed (other than scaling by the observed RWC), just the most
similar GMM curve was chosen, judged by the highest posterior probability. Also provided for reference are MGD curves with 1-, 2-, and 3-parameter fits. The 1-parameter MGD fits represent RWC-only fits, with $\mu = 3$ and $D_m = 1.18mm$ prescribed.

For the 2-parameter MGD fit, $D_m$ is calculated via Eq. 2 and $\mu = 3$ is prescribed. All the curves in Fig. 6 conserve total RWC as measured by the disdrometer.

Figure 6 shows a variety of observed DSDs from different locations and SST regimes. In these plots the size bins below $0.4\,\mathrm{mm}$ are greyed out to signify the disdrometers' insensitivity to these drop sizes. The discontinuities between size bins are noticeable in some panels at larger drop diameters, especially the fourth panel. The second panel is the most exponential distribution of the four shown, while the first panel shows a DSD that fits well with the MGD with $\mu = 3$ and a small $D_m$. The third panel shows a heavy tropical rainfall case with bimodal characteristics, as a high concentration of drops smaller than $D = 0.8mm$ is observed but significant concentrations of drops larger than $D = 2mm$ also exist. In this particular case the GMM-derived curves appear to provide the best fit but are still imperfect.

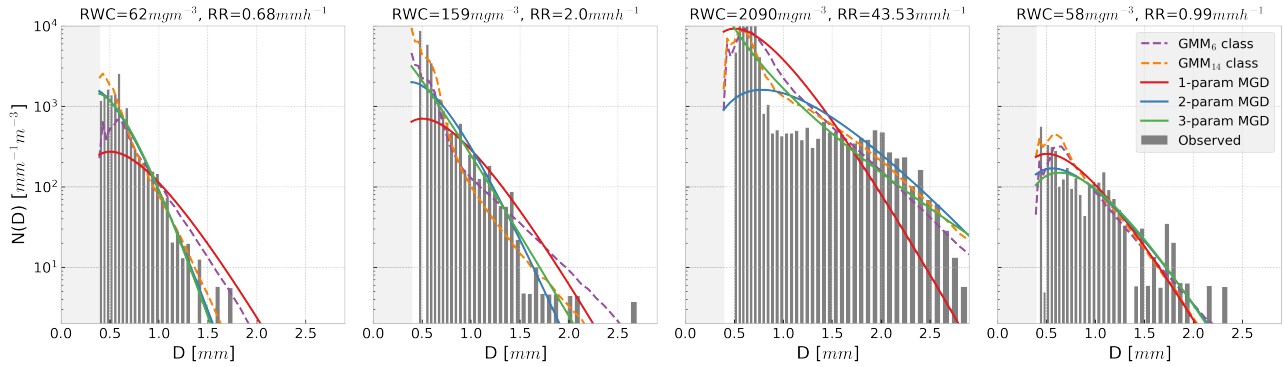

**Figure 6.** Each panel gives an OceanRAIN observed DSD, seen in the solid bars. Various fitted curves with identical RWCs are also given, including GMM-derived DSDs for $N_{GMM}$ of 6 and 14, and 3 MGD curves. For the 1-parameter MGD and GMM curves only RWC is provided, and for the 1- and 2-parameter MGD curves $\mu = 3$. The 2-parameter MGD has the calculated $D_m$ while the 3-parameter MGD (i.e. NG) also has the fitted $\mu$. Each data point is identified by its RWC and rain rate. The light gray shaded region indicates an area of no OceanRAIN sensitivity.

In contrast to the example plots of Fig. 6, Fig. 7 shows the mean GMM curves that arise from running GMM with a few different $N_{GMM}$ values. Again, this is from running GMM on the full disdrometer size bin data, with only the number of classes set a priori. For comparison, reference lines of NG distributions with sample $\mu$ and $D_m$ values are also given. Note that for each panel in Fig. 7, a majority of the GMM-derived DSDs feature more small drops than given by even the exponential ($\mu = 0$) line. In the simplest case with only two classes possible (first panel of Fig. 7), the DSD shape that best captures the majority of the OceanRAIN data set's variability (at least in terms of frequency of occurrence) is a shape that is more sloped than the exponential DSD, with many small drops and few large drops. This particular shape is common to all the GMM realizations, with even steeper curves found as GMM states are added. Indeed, the distributions produced by GMM seldom resemble a pure exponential DSD. It is an indication that a second shape parameter may be useful for describing oceanic DSDs, in line with the generalized gamma approach argued for by Thurai and Bringi (2018).

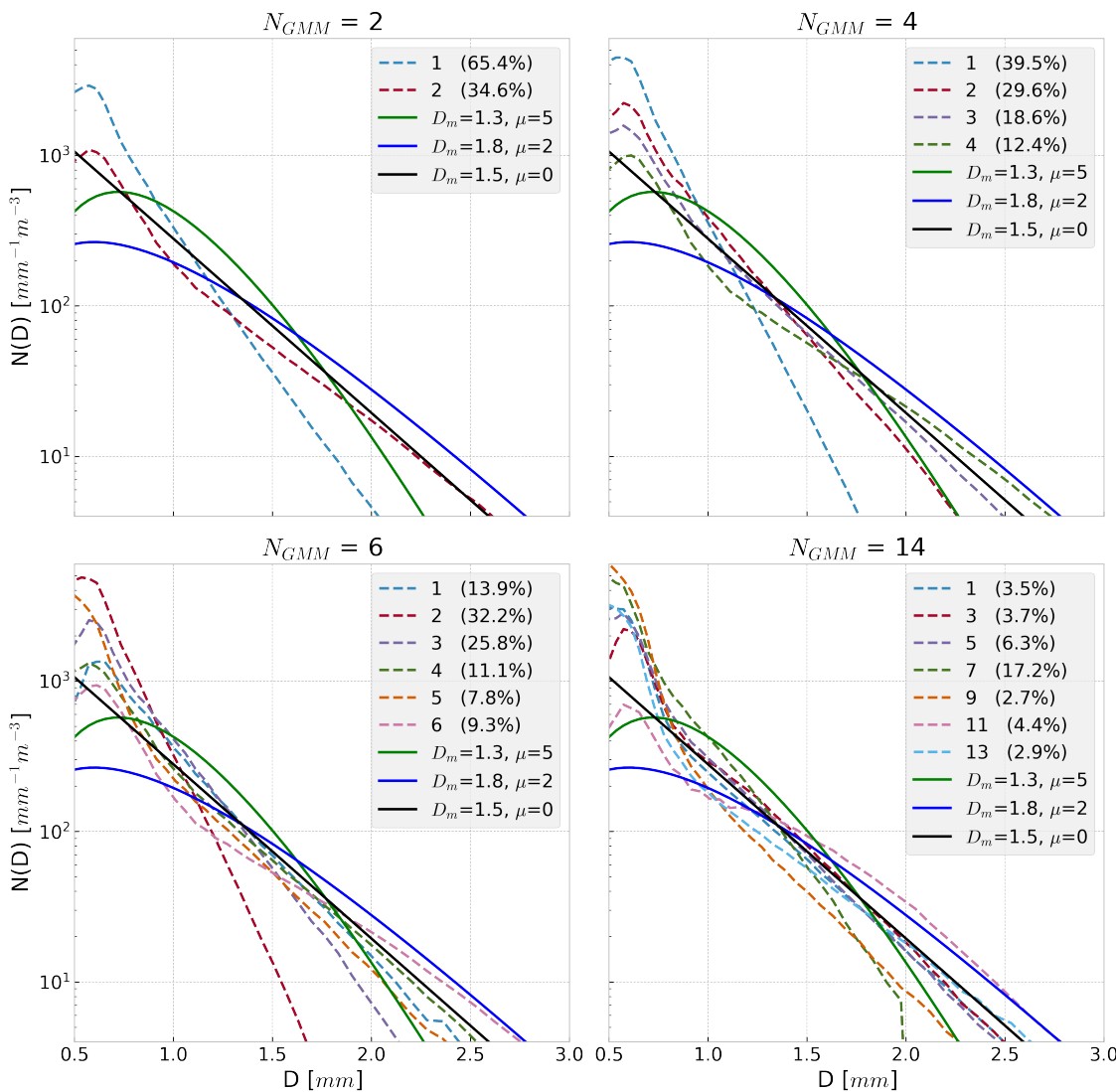

**Figure 7.** Panels show resultant DSDs for different GMM realizations (dashed lines) ranging from $N_{GMM}$ of 2 to 14. The last panel shows only odd numbered GMM states to reduce clutter. Each panel has an identical set of NG curves with different $\mu$ and $D_m$ values (solid lines) for the sake of comparison. All curves have equal RWC. The frequency of occurrence for each GMM shape is given in the legend.

It is noteworthy that most of the GMM states shown in Fig. 7 are not similar to the given NG curves across the full range of drop diameters. So while some of the GMM states are quite like a particular NG curve over part of the domain, it is rare to observe DSD shapes from individual minutes that resemble a 3-parameter MGD (i.e. NG) across the whole size domain. In many cases the GMM method prefers states with more steeply sloped DSDs and more small drops than the sample NG curves given. In fact, it takes higher values of $N_{GMM}$ (such as in Fig. 7 with $N_{GMM}$=14) before strongly peaked DSD shapes

reminiscent of NG with a large $\mu$ value emerge. In other words, DSDs featuring a strong peak near $D_m$, and for which an exponential is a poor approximation, are infrequent. This can also be seen in Fig. 3, as the PDF is relatively weak in the bottom left of that plot.

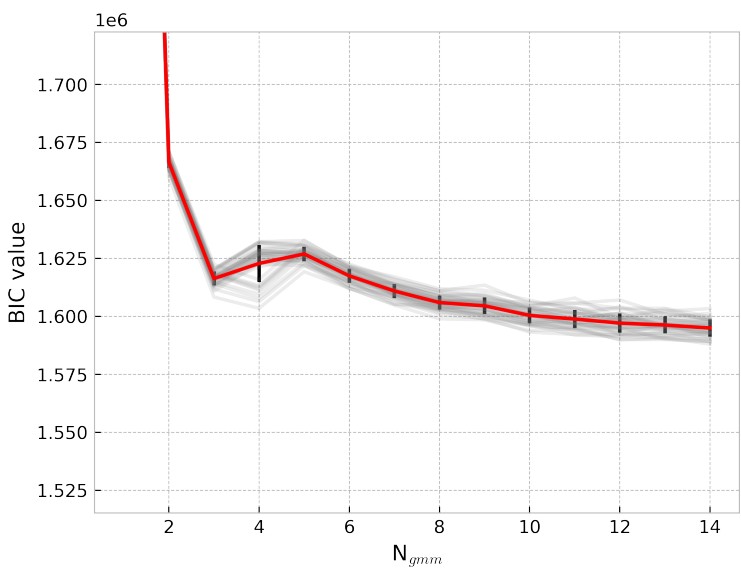

**Figure 8.** Bayesian Information Criterion (BIC) for different $N_{GMM}$ choices applied to OceanRAIN. The mean BIC is shown in red with the standard deviation in black. Gray lines indicate GMM tests with limited samples, each a randomly chosen subset making up a third of the total data set.

The GMM framework as applied to the DSD problem seems to offer the promise of finding a finite number of distinct shapes
with which global DSD variability can be described, a la Dolan et al. (2018), without constraining the type of shapes found. To investigate this, GMM was used in many iterations for randomly sampled subsets of the data to assess if an optimal number of states exist that describe the global shape variability. In this experiment $N_{GMM}$ was varied from 2 to 14. The Bayesian Information Criterion (Eq. 6) gauges whether addition of further states provides a better description of the data, shown in Fig. 8. BIC plateaus and continues a slight decrease for GMM states beyond about $N_{GMM} = 8$, indicating that there is no
singular set of GMM-derived DSD shapes that outperforms the others. Instead, oceanic DSD shape variability proves to be a true continuum that is not easily decomposed into a linear combination of a finite set of curves.

A corollary of the finding that a singular, optimal set of GMM-derived curves does not exist is that the observed DSD shapes do not display predictable regional patterns. The shapes observed are not distinct when normalized by RWC, whether considering the DSDs regionally or across SST regimes. The GMM-derived shapes are not tied to one region or another, a
finding that echoes Fig. 2. This is in contrast to some studies' success in pulling regional attributes out of large data sets via GMM without including location information, as was done here (Jones et al., 2019). The only area of OceanRAIN sampling that appears as distinct in the distribution of GMM states is from observations in stratocumulus regions, which are dominated

by the GMM states with steeply sloped DSD curves and a large number of small drops. Otherwise, the GMM states are not strongly tied to particular sampling regions. This tendency changes if DSD is not normalized by RWC, as RWC regimes are more tied to regional meteorology. But with respect to the retrieval problem, where it is convenient to separate the DSD shape from RWC as in Eq. 3, the GMM approach does not provide a magic bullet.

## 5 Radiative transfer impacts

An overlooked aspect of assuming a DSD a priori, or even just assuming the general shape of the DSD a priori, is that this will introduce forward model errors in retrievals and data assimilation. These errors can be strongly correlated across nearby frequencies and can thus cause systematic biases in variational systems (e.g. 1DVAR, 3DVAR) if not taken into account. An example of including this type of forward model error into a variational rainfall retrieval for GPM was presented by Duncan et al. (2018). Instead, the focus in this section is investigating the extent of forward model response inherent to variations in natural drop populations, without fitting a functional form to the observed drop counts. Because RWC or rain rate is usually the sought parameter from remote sensing retrievals, the results are separated along those lines.

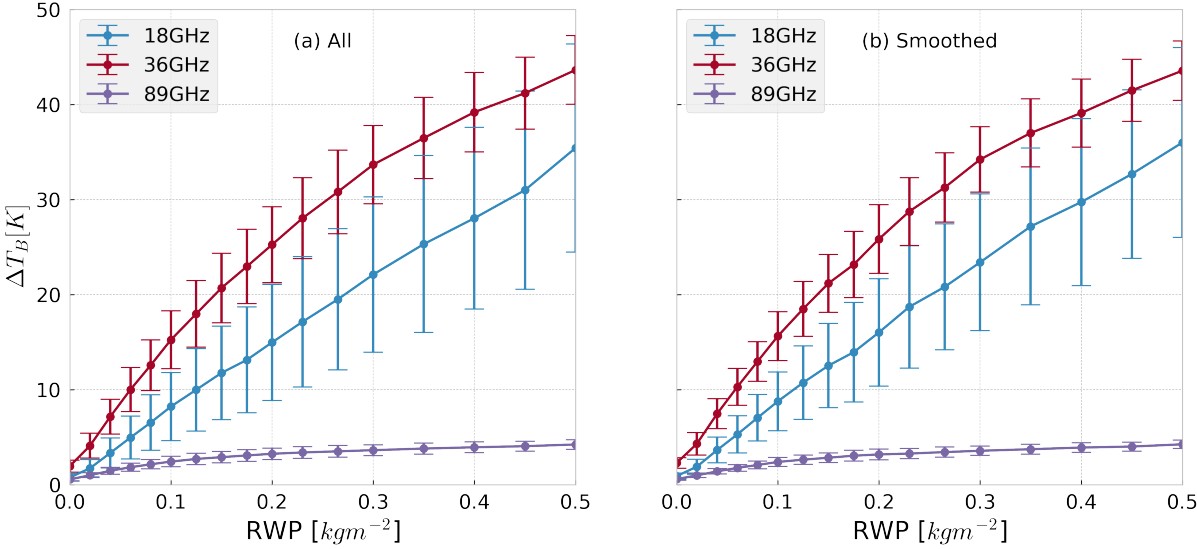

**Figure 9.** Simulated brightness temperatures ($T_B$) through a modeled atmosphere for warm rain, with a liquid cloud layer of $0.2\,\mathrm{kg\,m^{-2}}$ from 1 to $2\,\mathrm{km}$ altitude and rain in the lowest kilometer. The RWC in the rain layer and the DSD are directly from disdrometer observations and constant in the rain layer. Given are the means (dots) and standard deviations ($\pm 1\sigma$, shown as bars) of $\Delta T_B$ per rain water path (RWP) bin, where the difference in $T_B$ is defined relative to $RWP = 0$. The left panel (a) uses all OceanRAIN observations, the right panel (b) shows results when averaging over consecutive 6 minute observation windows to approximate a satellite footprint.

Forward model simulations of the radiative transfer were performed using the Atmospheric Radiative Transfer Simulator (ARTS) version 2.3 (Eriksson et al., 2011; Buehler et al., 2018). The ARTS model can handle custom particle size distributions

(such as observational size bin data) as well as prescribed DSDs such as the MGD. Thus with the full size bin data from OceanRAIN it is possible to simulate the interaction of radiation with drop populations without making any simplifications involving the drops' functional form. To approximate the impact on a sensor such as GMI on GPM, simulations were run using the GMI geometry and three GMI frequencies: 18.7, 36.64, and 89.0 GHz. Because the surface-based disdrometer data inherently lack vertical information, hydrometeor and humidity profiles need to be assumed. To avoid complications from inclusion of any ice scattering species, the setup is for warm rain: a $1\,\mathrm{km}$ rain layer defined by the RWC and DSD observed, with a $1\,\mathrm{km}$ liquid cloud layer of $0.2\,\mathrm{kg\,m^{-2}}$ above, characteristic of a raining warm cloud (Lebsock et al., 2008). Here we differentiate between cloud water and rainwater due to their different radiative characteristics, with the total liquid water path being the sum of the two. The surface properties and humidity profile are typical of a tropical scene, with the surface emissivity calculated using the Tool to Estimate Sea-Surface Emissivity from Microwaves to sub-Millimeter waves (TESSEM2), which is embedded in ARTS (Prigent et al., 2017). DSD properties are constant within the rain layer and the cloud layer is also homogeneous. Cloud droplets are monodisperse with diameter $15\,\mu\mathrm{m}$, whereas the rain drops are about two orders of magnitude larger in diameter, hence their differing scattering properties. Simulation code is available (Duncan, 2019).

Figure 9(a) shows the results of the GMI simulations using native disdrometer data, with rain water path (RWP) simply RWC vertically integrated over the $1\,\mathrm{km}$ rain layer, given in $\mathrm{kg\,m^{-1}}$. The change in top of atmosphere radiance in Kelvin, $\Delta T_B$, is defined relative to the non-raining case of $RWP = 0$ and for unpolarized radiation. With no mixed phase or ice phase hydrometeors in the atmospheric column, the three GMI channels chosen all exhibit a net increase in $T_B$. The $89\,\mathrm{GHz}$ shows little sensitivity to either DSD variability or an increase in RWP; its signal is mainly from cloud water emission, and the scattering signal from rain largely cancels out its emission signal from rain. In contrast, the lower frequency channels show large increases in $T_B$ with RWP as emission dominates and the cloud is more transparent, with the wide range of scattering response showing the strong dependence on drop size. The $18\,\mathrm{GHz}$ $T_B$ especially shows large variability for a given RWP, with the standard deviation of the $T_B$ response usually about half of the mean value. This is a significant error source for warm rain estimation, as the difference between a RWP of 0.2 and $0.3\,\mathrm{kg\,m^{-2}}$ would be difficult to distinguish using these frequencies alone due to the overlapping forward model error bounds.

To address the point-to-area issue of comparing OceanRAIN observations integrated every minute with those of a spaceborne passive microwave or radar footprint, which is $5\,\mathrm{km}$ in the best case, Fig. 9(b) shows a sample result if the disdrometer data are averaged in time. Averaging in time is performed because it approximates a spatial average, absent other observing points. Specifically, a nominal 16 minute window was used to average consecutive raining disdrometer measurements, in that a ship at $10\,\mathrm{kn}$ would take about 16 minutes to traverse $5\,\mathrm{km}$. Observations with zero rain rates were not included if the OceanRAIN points were discontinuous in time. Fig. 9(b) shows that the results are quite similar to the native disdrometer data used in panel (a), with standard deviations slightly smaller for lower $RWP$ values. The maximum forward model errors observed by a sensor such as GMI may not be markedly different than those presented with the time averaging performed, however most GMI channel footprints are larger than that of DPR.

Without needing to assume a model atmosphere, the variability of radar reflectivities can be simulated with the measured volume of drops alone and the T-matrix method (Klepp et al., 2018). Figure 10 gives the simulated radar reflectivity response

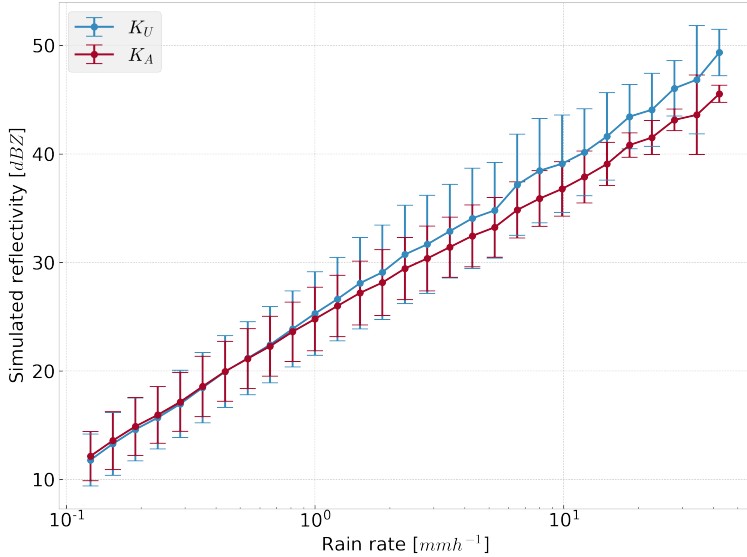

**Figure 10.** Simulated radar reflectivities at the two DPR frequencies, $K_U$ and $K_A$ bands, shown as means (dots) and standard deviations ($\pm 1\sigma$, shown as bars) binned by rain rate. The rain rate and the DSD are directly from OceanRAIN observations.

over a range of rain rates using the OceanRAIN observations. As with the passive sensor simulations, this demonstrates that DSD variability can cause significant differences in the radiative properties of a volume of drops even for equivalent rain rates or RWCs. As with Fig. 9, the range of scattering response is larger for the lower frequency channels, with $K_U$ showing greater variability in modeled reflectivity, as the specifics of the DSD determine whether the drops' scattering is wholly in the Rayleigh

regime or partly in the Mie regime. The $K_A$ band is less affected by DSD variations in both the passive and active simulations while scaling mostly linearly with increasing RWC or rain rate.

## 6   Discussion

### 6.1   Comparison with GPM

The discrepancies between OceanRAIN and GPM histograms of retrieved $D_m$ and $N_w$ (Figs. 4 and 5) deserve some discussion.

The distributions of $D_m$ and $N_w$ from disdrometer measurements are wider than those from GPM CORRA retrievals in most latitude bands, significantly so for $N_w$. GPM has limited sensitivity to small drops and lower number concentrations due to the minimum detectable signal from DPR, which may explain the small drops underestimated relative to the disdrometer measurements, especially in the Southern Ocean.

     The highly peaked GPM distributions of $N_w$, in stark contrast to OceanRAIN's much flatter $N_w$ distributions at all latitudes,

would appear to have two leading, plausible explanations. First, OceanRAIN is expected to observe more variability in the number of drops because it is a point measurement integrated over one minute and precipitation characteristics can vary widely

over multiple kilometers, whereas DPR has a $5\,\mathrm{km}$ footprint. Second, DSD retrieval from GPM is an under-constrained problem (more unknowns than information) despite the unique capabilities of DPR. While the altitude mismatch between surface-based disdrometers and the GPM data at a few hundred meters altitude may cause some systematic differences, say due to evaporation unseen by GPM, this does not explain the limited range of $N_w$ values retrieved by GPM. The strongly peaked $N_w$ distributions seem indicative of the significant influence of the a priori state on retrieval of $N_w$, in addition to the limited sensitivity to small number concentrations dictated by the instrument sensitivity of DPR. Future versions of the GPM algorithms may be able to make use of data such as in Fig. 2 to improve the a priori constraints guiding the retrievals.

## 6.2 Applicability of the modified gamma distribution

To examine the applicability of the MGD to observed ocean DSDs, we can compare the observed PDF (Fig. 3 and Fig. 11(a)) with the PDF of the same data but constrained by the NG fit (Fig. 11(c)). This is shown in Fig. 11(e), with sample NG curves given for extreme values of the shape parameter. The NG-derived PDF overestimates the frequency of points near the exponential line and displays less spread; blue areas indicate over-representation from the NG fit, red areas indicate under-representation from the NG fit. As with comparison between the PDF and NG curves in Fig. 3, this shows an underestimation of small drops at high number concentrations through virtue of being constrained by the NG fit.

To see if there is some latitudinal dependence within the overall OceanRAIN PDF, Fig. 11(b,d) divides the data into observations from high latitude (latitudes greater than $50°$) and tropical (latitudes less than $20°$) locations. It appears that whereas the NG with a shape parameter ranging roughly between $\mu = 0$ to $\mu = 3$ suffices for many tropical cases, high latitude observations are not always well represented by the 3-paramter MGD. For high latitude oceanic rainfall, Fig. 11(f) demonstrates that small drops are underestimated and medium drops overestimated if using the 3-parameter MGD.

One concern raised by the results of Fig. 11 is whether the use of the 3-parameter MGD, and its limited representation of the full PDF of drop sizes, can cause biases in modeled or retrieved rain rates. To examine this is quite straightforward, in that a size-dependent terminal velocity (Atlas and Ulbrich, 1977) can be assigned for drops of each size bin, with the rain rate calculated as the integral product of the velocity distribution and the third moment of $N(D)$. The calculated rain rate can then be compared between DSD representations. Using all OceanRAIN observations shown in Fig. 1 we calculated rain rates manually using the size bin data and assuming terminal velocities for all drops, allowing comparison of the rain rates that arise from the PDFs shown in Fig. 11 panels (a) and (c). The distributions resultant from the NG fit was found to result in a small mean overestimation of rain rates, by $0.06\,\mathrm{mm\,h^{-1}}$ or 1.9%. Using the same definitions as above, this underestimation was slightly less pronounced at high latitudes than for tropical latitudes, 1.5% versus 2.1%. This is due to underestimation of small drops by the NG fit, as small drops have lower terminal velocities than larger drops, and with RWC being equal this can have a minor impact on resultant fluxes of precipitation.

Much of the spread that exists in the full OceanRAIN PDF is due to the use of unsmoothed observational data that contain discontinuities between size bins and some degree of instrument error. It is clear, however, that much of the spectral power in Fig. 3 is not captured by any one NG curve. While the exponential line and $\mu = 3$ curves do a reasonable job at matching the PDF for larger drop sizes, the $\mu = -2$ curve performs much better for smaller diameters. This suggests that a 4-parameter

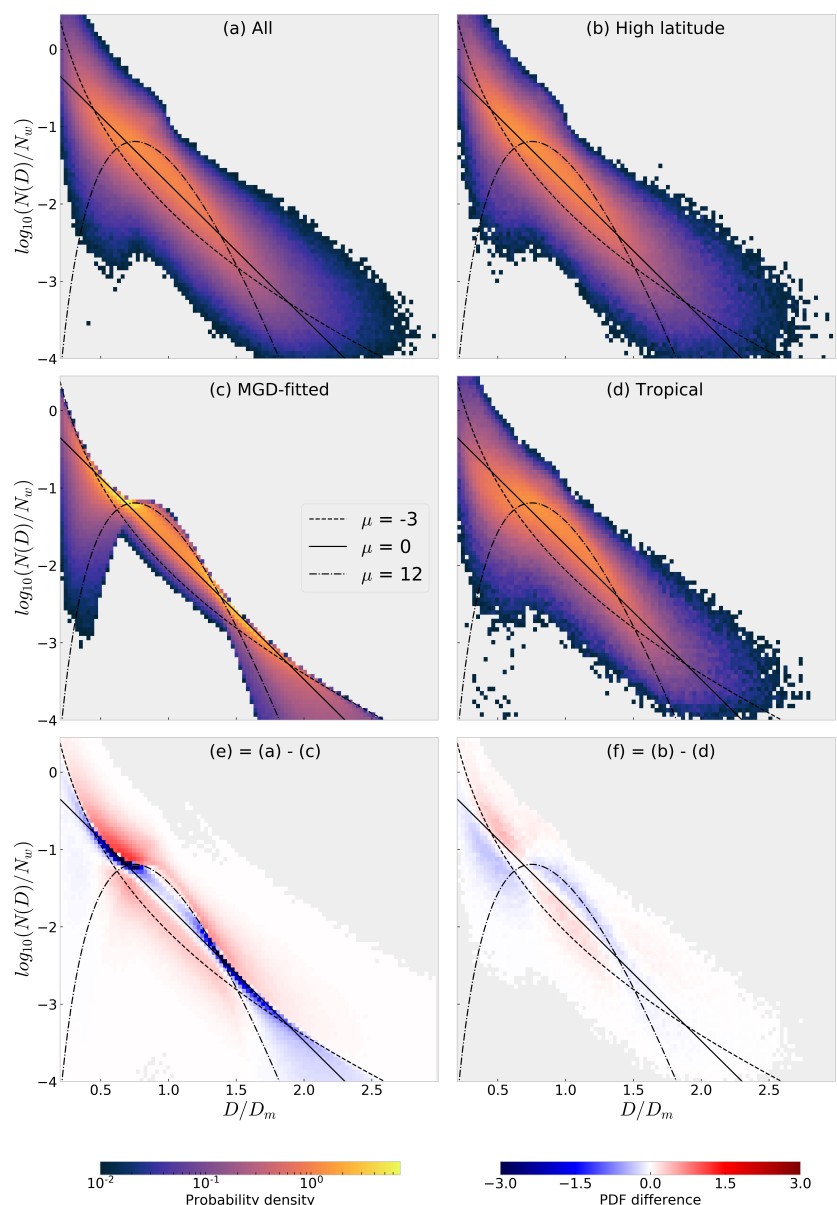

**Figure 11.** Panel (a) duplicates the result in Fig. 3. Panel (c) shows the data from (a) but after fitted to the NG distribution. The bottom left (e) shows the NG-fitted PDF subtracted from the full PDF. The right panels show OceanRAIN PDFs from high latitude (b) and tropical (d) latitudes, viz. $PDF_{>50°}$ and $PDF_{20°N-20°S}$, with their difference given in (f). Areas in gray indicate no data. The low and high $\mu$ curves given approximately bound the PDF space for the fitted data. Panels (a)-(d) share the same color scale and panels (e) and (f) also share the same anomaly color scale.

"generalized gamma" fit might be optimal for oceanic DSDs, a finding echoed in another recent study of disdrometer data (Thurai and Bringi, 2018). Use of the 3-parameter MGD can lead to some systematic biases in drop size representation as seen in Fig. 11(e). These biases can be regionally dependent, as shown by the higher number concentrations of small drops seen in high latitudes relative to the tropics, as seen in Fig. 11(f), and consistent with findings from Dolan et al. (2018).

## 7   Summary and conclusions

This study has investigated the variability of raindrop size distributions over the global oceans in a variety of contexts relevant to retrievals and atmospheric modeling. Methods to attach a functional form to raindrop populations vary, but have largely been predicated on limited land-based observations in the past. The OceanRAIN observation network of disdrometers provides an opportunity to move towards better understanding of global raindrop populations, with ramifications in aid of satellite retrievals and model parameterizations, which are necessarily global in scope.

The disdrometer data were shown to have limited dependence on latitude or SST (Fig. 2) when quantified using parameters of the normalized gamma distribution (Eq. 3). The mean and median of $D_m$ tend to vary within $0.1\,\mathrm{mm}$ across all latitudes, with $\pm\sigma$ of about $0.2\,\mathrm{mm}$. Most observations of $log_{10}(N_w)$ fall within 3.0 to 4.3 (Fig. 4), with a weak correlation exhibited between $N_w$ and SST (Fig. 2). These parameters from OceanRAIN were also compared to the leading estimates from a satellite platform (Fig. 5); comparisons with GPM matched relatively well for distributions of $D_m$ but less so for $N_w$. Both parameters appear to be too peaked from the GPM retrieval, likely a result of strong influence from that retrieval's a priori state as DSDs with approximately $D_m = 1.0mm$ and $log_{10}(N_w) = 3.9$ were frequent. The data sets exhibit similar spreads in the distributions of $D_m$, but the disdrometers show significantly more variability in $N_w$ than seen by GPM; the middle 90% of GPM $N_w$ retrievals fall within one order of magnitude, whereas the middle 90% of disdrometer observations span over 2 orders of magnitude. It was speculated that the GPM retrievals may be over-constrained, although it was expected that the point measurements of the disdrometer would display greater variability than those from satellite sources due to spatial and temporal considerations alone. Still, these results appear to demonstrate a systematic underestimation of number concentration variability within the GPM data set.

Usage of the normalized gamma distribution to describe all observed DSD behavior was questioned (Section 6.2), as it appears more applicable in the Tropics than for higher latitude populations. High latitude cases exhibit larger concentrations of small drops that are outside the state space specified by the 3-parameter MGD (Fig. 11). The 3-parameter MGD can cause systematic biases in rain rate estimation relative to using the observed size bin data, quantified to be a -2% error in the mean relative to total accumulation calculated from the disdrometers. This is a relatively small error for total accumulation because the drops that are most misrepresented by the normalized gamma formalism account for relatively little of the total mass flux; however, for about 3% of cases this is an error of $-0.5\,\mathrm{mm\,h^{-1}}$ or more, and can thus be significant. For many applications, an exponential DSD may be simpler and more appropriate than a NG distribution for oceanic rainfall (Fig. 3), but of course does not encapsulate the range of variability that exists, which may be better represented by a generalized gamma approach with four parameters (Thurai and Bringi, 2018).

Radiative properties of raindrop populations can vary rapidly for low frequency microwaves, manifest in Fig. 9 as the standard deviation magnitude is approximately half of the net radiative signal at $18\,\mathrm{GHz}$ but is much less at higher frequencies such as $89\,\mathrm{GHz}$. This is because the presence of a few larger drops can cause non-negligible Mie scattering that impacts the otherwise emission-dominated radiative signal and Rayleigh scattering from smaller drops, an effect that diminishes as

frequency increases. Fig. 10 also showed this effect, with lower frequencies exhibiting greater variability for a given RWC or rain rate due to observed DSD variability. Whereas the radiative variability is similar for light rain rates, modeled variability can be 2-3 times greater at $K_U$ rather than $K_A$ band. This observed $T_B$ variability caused by DSD variability is seen in both passive and active simulations. These ranges of forward model variability however represent a worst case scenario for satellite retrievals or data assimilation, as any skill in assuming or retrieving the DSD would shrink these ranges. This passive forward

model variability can even be viewed favorably, as it demonstrates sensitivity to the DSD at low microwave frequencies that may aid DSD retrievals. Simulations comparing forward model errors caused by using a GMM-derived or MGD state compared to the true DSD state showed that a high $N_{GMM}$ value was needed for the GMM states to outperform the 3-parameter MGD for forward model errors (not shown). This is in line with Fig. 8, but also indicative that it is hard for a single-moment scheme such as GMM to compete without having a large number of possible states.

This exploration of DSD shape "distinctiveness" was motivated by the remote sensing and modeling communities' need for simple but accurate parameterizations of rainwater's size distribution. For instance, if a region or meteorological regime tends to exhibit one or two DSD shapes, this simplifies a multidimensional problem considerably. The results, however, demonstrate that simple separation of DSD shapes by latitude and SST, or by other variables such as dewpoint temperature and RWC (not shown), does not significantly simplify the DSD problem. The limited spatiotemporal sampling of OceanRAIN meant that

further subdivision of regional data for seasonal shifts in DSD was not possible. The conclusion is then that global oceanic DSD variability, though more uniform than over land surfaces, is complex and not easily reduced to a single moment parameterization or a small set of possible shapes.

*Code availability.*  The code used for analysis is all available in the form of Jupyter notebooks via a Zenodo archive, found in the references.

*Author contributions.*  DD and PE conceived and designed the study, inspired by discussions with and the work of CK and DJ. DD performed

the analysis with aid from SP. DD wrote the manuscript and all authors contributed to its final form.

*Competing interests.*  The authors declare that they have no conflict of interest.

*Acknowledgements.* This study was funded with support from the Swedish National Space Agency, for which DD, PE, and SP are grateful. DJ is supported by a grant from the Natural Environment Research Council (grant NE/N018028/1), The North Atlantic Climate System Integrated Study (ACSIS).

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
