# Peer review of "On the distinctiveness of observed oceanic raindrop distributions"

_Atmospheric Chemistry and Physics, 2019_

## Referee Comment (RC1) · Anonymous Referee #1 · 18 Feb 2019

This is certainly a useful paper which is suitable for publication in Atmospheric Chemistry and Physics. However, there are some points which the authors may wish to consider prior to final publication, as follows:

Specific to the last sentence in the Abstract: Please see Munchak, et al., 2012: "Relationships between the Raindrop Size Distribution and Properties of the Environment and Clouds Inferred from TRMM", J. Climate, 25, 2963–2978. This is a relevant paper.

General: Note that MGD is sometimes refers to G-G, see for example, Petty, G. W., and Huang, W. The modified gamma size distribution applied to inhomogeneous and non-spherical particles: Key relationships and conversions. J. Atmos. Sci. 2011: Vol. 68, pp. 1460–1473.

[Figure]

Page 2, line 5: Refer to Testud et al here.

Page 3, line 31: What is the size resolution of the ODM?

Page 4, line 4: Please clarify "impacts of turbulence"

Page 4, para 3: It's not clear why the accuracy should depend on oceanic or continental cases. Bumke and Seltman showed that the average DSD shapes were similar in coastal and continental locations. They used the scaling of Sempere-Torres et al to fit the coastal and continental DSDs and found an invariant shape. Perhaps this should also be mentioned here.

Page 7, end: Recommend plotting Nw vs Dm on semi-log scale to further illustrate the inverse correlation between the two, for ODM and GPM.

Page 9, line 5: It's not clear why mu=2 fits the mode of the normalized DSD shape in fig. 3 especially for D/Dm<0.5. In fact mu=-2 appears to be better.

Page 9, line 8: what method? Moments-based, MLE...

Page 9, line 15: Please clarify this range for mu for tropical cases. The mu estimate is very sensitive to the shape of the small drop end. It is not clear what the resolution and accuracy are for the ODM at tiny-small sizes. No independent evaluation of the ODM accuracy is given for the small drop end. Hence, caveats are recommended when statements regarding mu ranges are given.

Page 10, line 2: What is meant by "moments" in this context? Are you referring to the moment-based estimation of mu?

Page 11, line 6: use of "power" is not conventional...please use another descriptor.

Page 11, line 15: the moments fit should be explained earlier.

Page 11, line 15: Units for RWC are not clear....seems like mg/m^3. No discussion of fig.5 ? Except for second panel, the remaining GMM are not close to the measured

N(D).

Page 12, line 2: At this point, recommend that the 4-parameter gamma be used to illustrate that 2 shape parameters are needed.

Page 17, line 5: The conclusion is based on the assumption that the resolution and accuracy of N(D) from ODM for small drops is well-established but this has not been demonstrated. The use of total accumulation is not the only criteria by which one justifies the use of the MGD. The width of the mass spectrum is considerably increased when the small drop end is accurate. From cloud physics viewpoint the N(D) is a result of various microphysical processes that are controlled by the low order moments including M0. The GPM algorithms use path integrated attenuation as a constraint...the attenuation is also sensitive to the small drop end especially at Ka-band.

Page 17, line 29: What about stratiform vs convective vs shallow rain types?
* * *

---

## Referee Comment (RC2) · Anonymous Referee #2 · 3 Mar 2019

The paper presents many interesting results using a wide variety of relevant and novel data sources. The aim and motivation of the paper are well-justified - the paper addresses the need to physically understand rain variability from observations and to accurately retrieve this information with the currently-available suite of satellite measurements. I am highly supportive of the paper. However I find many areas that need improvement because the way things are written currently (vaguely, with results confused with discussion) leave many statements open to interpretation.

The subject matter of this paper is worthy of publishing, but the paper first needs to be revised in several ways:

1. Reorganize to separate Data Information / Tools (the datasets, their intricacies, and the models you ran them through) from Results (what your methods produced)

from Discussion (as a logical conclusion of the results shown, what do the results mean?; what are the implication of results based on the methods you used?). Right now, methods, results, and discussion are mixed throughout each paragraph. This often means the authors often repeat themselves. The main objectives of the paper and main results of the paper are obscured by this mixture of results + data details + discussion in each paragraph, and from paragraph-to-paragraph. There is not enough logical order within each paragraph or from paragraph to paragraph. Right now, the paper is separated thematically by the data used, which also serves a purpose but is tiresome for the reader. As written currently, the authors don't leave a well-defined trail for the reader to follow, instead they loop back and forth over their tracks several times and lead the reader to wonder where the story is going, where has it been already, and why. Consider implementing a roadmap of Background, Motivation, Objectives, Data, Methods, Results (just the facts - just state exactly what you did and present the figures objectively), Discussion (interpret the results, draw nuanced conclusions, and comment on the significance/implication of the results for other applications), and Summary (tell us how you met the challenge stated, hopefully clearly, in the Motivation/Objectives. Progress from section to section, don't repeat yourself throughout.

a. The most egregious instances of mixing discussion with results are in Sec. 3. The beginnings of many of these paragraphs should be deleted b/c they are redundant, or the methods are restated. The last sentences of many of these paragraphs appear to opinions of the authors and should be separated from the results and kept in a discussion section. Ex - last sentence of 3.1 is not a logical conclusion from data shown, it's based on the experience of the authors and their interpretation.

2. Rewrite/Revise each sentence for literal truth. Each statement should be made to be physically true, word for word. So many of the sentences in this paper contain vague, non-specific, perhaps physically impossible language that is abstract and doesn't really teach the reader anything. The authors need to more specific detail in several places so that their results are digestible, understandable, and reproducible. In many cases,

I did not understand what the authors even did in terms of methods, how they drew the conclusions stated, what certain plots really showed, or how to draw conclusions from these plots. In many cases, problems/issues are alluded to (e.g. uncertainty, constraints, sensitivity), but the specifics or causes of these problems are withheld so the reader is left with suspense - what could the problems really be and why are they there? We don't know, because the authors allude to issues without explaining them. Examples of vague language that doesn't really explain anything physical:

a. GMM easily generalized to a wide variety of data distributions * What is meant by easy? What does generalize mean, physically? How wide? Of what variety? What are data distributions, in this context?

b. Underpinning OceanRAIN is the ODM-470 optical disdrometer * Does a disdrometer literally underpin (lie beneath, support, substitute for weaker material) the data network? No. These instruments are used to collected the data amassed in this network.

c. Limited sensitivity of DPR to small drops * How sensitive? How small of drops can be seen?

d. Has a more bimodal distribution * How do you quantify what is more or less bimodal? The PDF either exhibits two peaks in frequency of occurrence, or not.

e. Commonly occuring, common forms, commonly used DSD, and other uses of common * Used by whom? How many times? What makes it common? How can a DSD be commonly used? All uses of common should be replaced by a descriptor that is physically-founded and/or citable, as in which studies/models used it, or how many?

f. Data that contain discontinuities between size bins and some degree of instrument error * What discontinuities? How is this quantified? * What degree of instrument error? And why?

g. Unmolested observations * NO - delete/revise for something more physical and less

politically/emotionally charged. h. Magic bullet

* NO - delete/revise for something more physical and less politically/emotionally charged. i. "Robust", or even worse "very robust"

* Search all instances of this word and replace with a physically-based and specific metric. Error, uncertainty, skill, accuracy... what do you mean by robust? How is that quantified or assessed?

The paper could be shortened, simplified, and made to be more digestible and impactful if the sentences were simplified. The authors should carefully edit the manuscript to just state the facts, use simple sentence structure, and use specific numerical details (above 2.5 mm, greater) instead of subjective, nuanced, literary adjectives or descriptors. Examples of overly verbose, passive voice, unspecific, and incomprehensible statements often contained in this manuscript: j. OceanRAIN observes a wider distribution of Dm that is "most noticeably distinct from GPM results for small drops". * You lost me with the phrase in quotes... what does this mean? How small? How do you quantify what is "most noticeable distinct"? Are there more physical or statistically significant ways you could try to prove your point here?

k. Functional forms, models, fits... * Pick one and stick to it? Are they different?

l. Underconstrained, unconstrained * By what metrics? How is this defined? It's mentioned several times that retrievals are underconstrained/unconstrained, and this is identified as an issue. However, it's not clear what the issue really is. What is missing from the constraints?

m. Have the correct RWC * Correct to: conserve total LWC compared to observations

n. not particularly Gaussian * Particularly? It is either is or is not Gaussian

o. This can then be compared between DSD representations * I have no idea what this means

p. Warmer ocean surfaces witness greater densities of drops * Since oceans do not have eyes or consciousness, it might simply be the case that rain exhibits higher number concentrations when it develops and falls over water with warmer SST.

q. May or may not make much difference. * Compared to what? What differences are you referring to? What would change, or not?

r. Bulk radiative, bulk properties, bulk parameters * What is meant by bulk here? What does it summarize? It seems like a filler word that misses the point, if there is one. Vertically integrated? Averaged? From 1-m depth, over the lower km? Use physical descriptions instead of vague adjectives that are open to interpretation.

s. Peaked distribution, smaller drops, fewer drops, higher, lower, more, less, tail, maxima, very weak, very strong, large range, lower resolution, relatively similar, overall, generally, overall behavior, extreme values, less spread, slightly less pronounced, scant, much of, underrepresentation, markedly different, not particularly, not especially distinct, and any instance of slight, almost, barely * All of these extraneous, subjective, hand-wavy descriptions need to be made more physical/numerical or deleted: Mode = most frequent value, was it the actual maximum value? How much higher? How small? What diameter ranges? How extreme? How low? Very is a filler word. Just quantify what you mean. The data should speak for themselves. Overall and generally are also meaningless filler words. Quantify your analysis to prove your point. Use numbers to justify your interpretation. * What does underrepresentation or overrepresentation mean? See discussion of Fig. 3 on Page 9-10.

t. Posterior * Replace all instances of this with something more plain-language or physical. Not immediately apparent how an after the fact (?) step is executed.

u. Represented by this formalism * ?

3. I'm unsure about the title after reading the paper. The authors conclude that there are several forms of DSD shape/solution, all of which overlap a bit, and their large

datasets include instances of DSD that oscillate around these forms (as shown by the machine learning). Then they make conclusions about how well in situ observations of DSDs are represented with current satellite DSD retrievals (sensitivity problems appear to limit the capabilities of the satellites), and how much scatter should exist in satellite retrieved Zh and Tb given the observed DSD (quite a bit, and it varies based on frequency). The paper really doesn't separate out "raindrop regimes"... which would indicate to me to be physical distinctions like monsoon phases, convective v. stratiform, different meteorological conditions, etc. Perhaps the impact and message of the paper is best summarized as: "On the distinctiveness of oceanic raindrop distributions in observations and satellite measurements"

4. In intro: "A lack of globally representative DSD"... do we lack that? What is lacking, specifically? Are you saying that we need to find a single DSD that represents all DSD across the globe (implied by the way this is written), or that we need a representative dataset? Unclear. 5. The authors don't give enough credit to the oceanic-ish DSD measurements that are still being reported at Manus Island and Kwajalein Atoll. These are not representative of other DSD in other regions, but they do contribute something important; not all the DSD studies have been confined to land or coast (Thompson et al. 2015 and 2018, and this data is also used in Dolan et al. 2018).

6. The authors make up a quantity of RWC and RWP, when it seems that literature to-date uses LWC and LWP to mean the same thing. Paper should conform to these precedents instead of inventing or using their own terminology. It's clear in the paper that only rain cases are used, not ice- or mixed-phase precip, at least at the ground.

7. Fig. 2 - each of the plots must be spaced farther apart in the Latitude dimension so that the distributions can be viewed as distinct. The plots run into each other and cannot be distinguished one from the other. Therefore, the point of this plot is lost. Also, It's not clear what the latitude averaging bin width was here. It's centered 20deg apart - is that the center value +/- 10 deg in each direction? Explain in text and caption

8. The units of Nw need to be checked - it's often discussed in the text as Nw when it's really logNw that is plotted. The parentheses should be adjusted to explain the units properly. log10(Nw [mm-1 mm-3]). Log10Nw doesn't have the same units as Nw itself.

9. Each of the box and whisker plots need a legend that explain the box width, lines dots, cross hairs (what are the percentiles, etc.). It's too much work to read the caption and try to interpret the plot at the same time.

10. Add a plot of the map of observations used, so that your Fig. 1 and comparisons between latitude bands can be better understood. The text mentions that more or less data was collected in certain regions, but the reader can't deduce that because no plot of the data extent is shown.

11. Fig. 4 is just about incomprehensible. The conclusions about this figure do not appear to stem logically from the plot, and the plot itself is confusing. If you are making a comparison between models and obs, that is one thing and can be shown a certain way. Then if you also want to compare latitude bands, you should show the PDFs from those latitude bands first before you plot the difference. The difference here doesn't really make sense either because it doesn't seem physical to subtract a tropical PDF from a higher lat PDF; the result doesn't really have any physical significance (or at least the significance is not well-explained or logical). Suggest deleting Fig. 4 and showing the real tropical and high latitude PDFs. Then see if your stated conclusions are supported by the real data. The entire discussion of Fig. 3-4 needs to be carefully curated to be sure that the interpretations of the authors are readily apparent from the plots shown; I couldn't really see how they made all the conclusions that were stated. Particularly the regional dependence, but also the under vs. over "representation" issue. Fig. 4 needs titles to distinguish them and explain more about what is shown.

12. I kept forgetting what GMM and MGD meant. . . these acronyms are used throughout and their physical significance became lost several pages in. Suggest using plain language. See. Fig. 5 - it's not clear what the significance of the acronyms are for the

purposes of showing the data.

13. Typesetting should be improved a. The lack of line numbers on each line makes it difficult for the reviewer to suggest particular comments and corrections in each case. The authors should number each line in the future. b. The paragraphs would also be easier to distinguish if their indentation was larger. It kind of all runs together c. The references within parentheses are wrong. For example this is typed often: (also used by Duncan et al. (2019)) < see double parentheses at end here. d. Most journals forbid sentences to begin with the name of the variable and will insert words to rectify this situation, whether or not it makes physical sense. The authors are encouraged to fix this the way they want before it gets changed for them (see paragraph above equations on page 2). Completing these sentences also will help ground them in concrete, physical terms: e.g. correcting to be: "The values of N0 and NW..."

14. Equation (3) is really 3 equations... separate them?

15. Nw is normalized by LWC. make this clear in its discussion on page 2 and elsewhere.

16. Petkovic et al. 2018 reference page 2 - this is not the definitive nor the first paper to study these things. Reference prior work more appropriately to justify this statement - other canonical papers?

17. References are made to OceanRAIN-M ... what is the -M??? Not defined.

18. The ODM470 doesn't sense drops smaller than 0.4 mm because the voltages are too noisy below this inferred level (the sensitivity is incorrectly stated as 0.3 in the manuscript). From Klepp 2018: "The first 12 bins ranging from 0.04 to 0.36 mm in size are not recorded because they are prone to contain artificial signals caused by ship vibration." This is a significant downside to the instrument that needs to be explicity stated in the manuscript. A large portion of the DSD spectrum is contained below this level (Thompson et al. 2015, Thurai et al. 2018). Also, the ODM assumes a

fall velocity and does a vector mean of the fall velocity + wind through sensor. This is an approximation and, in my interpretation of the ODM methods, a potential source of error in the computed number density.

19. "Is an issue faced by all disdrometers" -> ya, but it's the worst for ODM compared to other disdrometers currently used. See comment above.

20. Minimize effects of turbulence -> minimize under-catchment of drops and minimize the effect of drop splashing against the sensor from multiple directions. Turbulence will still exist. It's just that the orientation of the instrument attempts to catch as many drops as it can.

21. Why would the disdrometer NOT "show no difference in accuracy between oceanic and continental cases"? Why would differences occur? It should work the same, except that while at sea, it has to reorient into the wind and a vector mean calculation is done to assume fall velocity+wind through the sensor, which could be susceptible to flow distortion.

22. Introduction of the "Combined Data" is confusing. The heading of that subsection is also confusing. What does it mean?

23. State the different vertical resolutions of the two DPR frequencies. It's mentioned that they differ, but not what they actually are.

24. The specifications of the GMI and DPR (geometry, resolution) are scattered about the last paragraph of 2.2. Make it more concise.

25. "Ground-based data from Ocean-RAIN-M" ⸺ Ocean rain is over ocean! Not land.

26. How is clutter classified in satellite retrievals? Sec. 2.2. Seems like it would impact your analysis.

27. Provide citation for GMM upfront at its first introduction

28. 90,000 points seems REALLY low compared to the size and length of time of the

OceanRAIN dataset. Is this correct? I've collected ODM data over a month and gotten more than 40,000 points of usable data.

29. What are the size ranges of the 60 size bins? And what are their approximate spacing?

30. In several places, the authors assert that assumptions and fits haven't been used on DSD obs, but then they say that the data are normalized and that a "nominal shape parameter is assumed" .... This seems contradictory. And the authors fail to mention how the data are normalized - by liquid water content? Similar to Nw? how was the parameter chose?

31. Provide citation of BIC when introduced

32. The authors make claims that they are testing for differences in DSD in different regions or locations, but really they only separated data into very wide latitude bands (Fig. 1, Fig. 2). This distinction, and the implication of the real results, need to be me explicit so as not to overstate the significance or conclusions. Search all uses of "region" and "location" to determine whether you actually mean "latitude"

33. Similarly, the authors make mention of a stratocumulus region, but have not proven where this is or justified their interpretation of that based on a map of the data - which should be added to justify statements made throughout the analysis in Sec. 3

34. The differences in DSD based on SST seem readily explainable from the Clausius Clapeyron equation - warmer SST leads to higher saturation vapor pressure of air, so moisture content can be higher. – see Fig. 1 and Fig. 2. Nw is directly proportional to LWC (your eq. 3), so result of Fig. 1 lower right is awesome but perhaps not surprising.

35. "Blue" dots in Fig. 2 don't show up. Looks black.

36. What are the bin sizes of latitude and SST used to create this plots?

37. Captions and discussions of FIg. 2 say Nw but you plotted log10Nw

38. The satellite is not sensitive to smallest drops, and probably also not to the lowest number concentrations. This seems to explain some of the differences in flat vs. peaked DSD shape in FIg. 2.

39. Evaporation below the lowest altitude of GPM would impact the smallest drops first (see Matthew Kumjian et al. papers with observations and models). So, this effect, if it's occurring, would eliminate the small drop portion of the spectrum first, most likely. Since small drops usually present in large number concentrations (Thompson et al. 2015, Dolan et al. 2018), this might mean that the satellite also misses large number concentration DSDs as a result of missing the smallest drops.

40. The exponential DSD is the most commonly occuring, which is why large combined or averaged datasets of DSD often exhibit a shape of this kind (explained in Bringi and Chandrasekar 2001). However, this book also states why you wouldn't expect high-frequency observations of DSD (such as from 1-min observations) to look exponential. The Marshall Palmer was also based on stratiform, steady, UK rain... so again, it's most representative of this steady-state, averaged, stratiform, weakly-forced rain DSD.

41. It is stated on page 10 that rain rate is 3rd moment of DSD; it's actually 3.67th moment. LWC is 3rd moment (Bringi and Chandrasekar 2001).

42. "Result in a small overestimation of rain rates by 0.06 mm/hr or 1.9%" - unclear how this was calculated? At all rain rates? Or overall error? I'm confused how the error in mm/hr can be just one number?

43. Raw observations - it's not really raw. You selected certain sizes, the data have been converted from voltages at fractions of a second to DSD parameters based on A LOT of assumptions contained in the Klepp papers. They are not raw, but they are observations you described already in methods. Just don't use a new qualifier to describe them a different way.

44. How were the "random samples" and "random sampling" and "randomly sampled

subsets" performed? Chunks of data taken at random? How much? Explain.

45. Page 11: "curve was chosen"... I just see lots of curves on Fig. 5. How was it chosen?

46. Last sentence of 4.1 - was this also seen by Dolan et al. 2018?

47. Text at top of Page 12: it seems that you are trying to say that all GMM output DSD fits depart from the exponential form, so that observed mu almost never equals zero.

48. FIg. 6 - label legend better in terms of GMM and GMD parameters (and explain in plain language, it's not clear here). In caption, differentiate solid and dashed lines. Water content should be LWC. Keep consistent terminology/abbreviation throughout.

49. Page 13: rare to have -> rare to observe; observations -> minutes

50. Last few sentences before Fig. 7 does not make any sense; overly literary and wordy. And it's not clear how these conclusions can be drawn from what is shown.

51. It's not clear how interpretation of BIC in Fig. 7 leads to the conclusion made by the authors. I'm totally confused how there can be a singular set... seems like an oxymoron and I don't know what this is intended to mean. It actually appears like 2 modes describe most of the variability (highest BIC) and then Ngmm from 3-6 does fairly and equally well.

52. The last paragraph of Sec 4 is very problematic and hard to follow, and hard to deduce the conclusion from the results shown. The claims appear unsubstantiated as written. I don't know what "decoupled from RWC" . . . there was no regional comparison performed. . . use of "not particularly" and " not especially distinct" are handwavy and should be replaced by quantifiable and specific terms.

53. What are "variational" systems? Page 14

54. "Water content" is stated. . . but else where its RWC. stick with consistent terminology/abbreviation once defined, otherwise it sounds like you are introducing a new

concept or field.

55. Page 14: "was used to perform model simulations" . . . of what? Incomplete sentence.

56. Page 14: what are the custom distributions, or habits? Mentioned but not explained sufficiently.

57. Unclear what g m-3 cloud looks like or what this number is supposed to represent.

58. Define TESSEM2

59. How was the change in TB computed? From level to level or from certain simulations? The description of Fig. 8 and these simulations is very confusing. I don't really know what you are trying to do or how you are doing it. Clarify. WHat are the units of TB here? What about units of RWP? Units need to be stated in the text. I don't understand what impacts are being tested here , "The 89 GHz shows little impact" . . . impact of what on what?

60. What do you mean by"largely cancels out its emission signal?" Page 14

61. What are the sizes of cloud droplets and rain droplets that produce these stated differences in emission vs. scattering?

62. Fig 8 and Fig. 9 and other figures where error bars (of some kind?) are shown - need to define on legend what the errors bars and whiskers and dots really mean.

63. Add titles to Fig 8 so that you can tell from looking at the plot what the differences between a and b are.

64. Fig. 8 discussion: "net response" should be "mean value" according to your plot. Be specific about what you are discussing. I'm confused what "response" means in this context.

65. The averaging window of 6 min seems too short to prove your point. Assume

that ships transit at 10 kt (nautical miles per hour) - a good assumption, or you can test it with all the meta data in the ODM dataset. Anyway, figure out how long in time you need to average in order to approximate a 5 km or degree or whatever distance spanned by a single pixel of satellite data

66. "Should not be markedly different." . . .. If what? Compared to what? A comparison or warning seems to be made, but it's not clear what it is.

67. "Non-raining points" – do you mean that you did not average zero values? Unclear

68. Fig. 9 - why aren't data shown above 40 mm/h? Surely they exist in ODM?

69. The author's conclusion that the error is way bigger for the blue line is not well-supported. The errors are only different in blue v. red at the very highest rain rates (above 10 mm/hr). Consider revising interpretations for clarity / consistency.

70. Page 16 "largely been predicated on limited land-based observations in the past" - including some island observations from Manus, Gan, and Kwajalein.

71. Page 16: here it is stated that DSD have limited dependence exists on SST, but in presentation of Fig. 1 you remarked that there appeared to be an SST dependence. Inconsistent.

72. Page 16 in several places: datasets "observe" -> exhibit. Not literally true.

73. Summary and Conclusions: your latitude bands are very wide and may inhibit your ability to draw conclusions about DSD variability that could be present in regimes such as the monsoon, ITCZ and SPCZ, subtropical highs, western boundary currents that are really warm, cold current regions, etc.

74. "Function to encapsulate" and "appears more applicable" – > unclear. Revise.

75. Line 3 page 17: spatial and temporal considerations

76. Unclear what this sentence means "Its use can cause systematic. . ..

[Figure]

77. Sentence with "for about 3% of cases ..." = run-on sentence. Revise to simplify it.

78. Page 17 lines 12-24 make very little sense. Revise for clarity a. Uncertainty = standard deviation? Width of distribution? means? b. What is "radiative uncertainty"? c. How "rapidly" d. What is meant by low vs. high frequencies? e. "Half the radiative signal" - do you mean the mean? f. What is "true for passive and active simulations" ?? uncler based on what has been shown or discussed g. Unclear why "retrieving the DSD would shrink these ranges" unsupported from what is shown or explained, or maybe it just doesn't make sense h. Lines 19-24 don't really make any sense. Revise. Unclear. i. It's confusing that you bring up dewpoint temp and RWC on line 28 because you didn't discuss Td anwhwere else but you did use RWC. j. What is meant by "limited spatiotemporal sampling of OceanRAIN"? – Please include a map in the revised manuscript k. What references can you provide to justify why DSDs are more uniform over land. They seem to exhibit more modes over land because hail and mixed-phase and very intense convective microphysical processes can occur. Unclear from what is written.

I tried to understand what the authors did, which is actually a lot of work and very interesting.

The paper synthesizes a wide range of data types:

This paper presents DSD data from 90,000 individual minutes of rain sampled throughout oceans. The data were collected with the ODM-470 disdrometer mounted on 9 research ships between 2010 to some time in the present (the cut off date of this analysis was not specifically stated). The study also makes use of satellite measurements and derived quantities at each of the times and approximate locations of the in-situ DSD data. The satellite data used in this study include the vertical profile of DSD parameters: $N_w$ (lower - how low?? - vertical resolution) and $D_m$ (250 m vertical resolution), while the shape parameter mu is assumed to equal 2. These DSD parameters are retrieved along the DPR vertical profile of ... [what DPR values are of interest??

reflectivity, Zh? The retrieved field of interest from DPR is not specifically stated but should be]... at 5 km horizontal resolution at two frequencies (Ka, Ku), and also the GMI brightness temperature, Tb, at a [wider?? Not specifically stated??] horizontal dimension at several frequencies (10-190 GHz). This set of data is from the "GPM Combined" dataset.

The analysis of these data is two-fold.

1) For both the in-situ and satellite data of DSD, compare these observations to functional fits and shapes derived from (a) numerical solutions to physical equations (b) machine learning models. 2) Using only the in-situ data, investigate how the DSD and the DSD's liquid water content/path impact two remote sensing quantities (brightness temp and radar reflectivity). The mean dependence of the satellite quantities on these DSD properties is assessed, as well as the variance and standard deviation for different bins of LWC/LWP/drop size. This task is performed at different frequencies used operationally in satellite rain retrievals passive microwave and radar systems. By examining different frequencies, the authors are able to compare the sensitivity/variability of satellite measurements to certain DSD properties as a function of satellite frequency. The standard deviation/variance in modeled satellite retrievals resulting from initialization with the full dataset of DSD is assessed, and referred to as the forward model error.

---

## Author Comment (AC1) · 29 Apr 2019

Attached is a combined PDF of the full response to reviewers and the tracked changes document.

Please also note the supplement to this comment:
https://www.atmos-chem-phys-discuss.net/acp-2019-34/acp-2019-34-AC1-supplement.pdf

---

## Author Response (AR1)

**Response to Reviewers, ACP-2019-34**

David Ian Duncan, Patrick Eriksson, Simon Pfreundschuh, Christian Klepp, and Daniel C. Jones

**General response**

Thanks to both reviewers for the in depth and thoughtful comments offered on ways to improve this manuscript. In accordance with suggestions from both reviewers, the follow major additions have been made: a figure showing OceanRAIN sampling (Fig. 1), a table with OceanRAIN size bins (Table 1), a figure with OceanRAIN and GPM $N_w$ vs. $D_m$ (Fig. 3), more panels to elucidate the PDF differences in the normalized gamma space (now Fig. 11), and a penultimate section to separate better the results from discussion. The title has also been modified. The histograms of GPM and OceanRAIN data separated by latitude are also now presented in a different plot format due to reviewer and public comments, and all GPM data used for analysis have been updated from V05 to V06 to make the analysis as up to date as possible. The reviewers' comments have led to refinement of the manuscript's language in numerous places throughout the paper.

In the following, the reviewers' comments are given in italics, with specific responses following each and any additional text added to the manuscript then given in quotes. Changes to the manuscript are also visible in the 'tracked changes' document uploaded alongside this response.

**Reviewer 1**

*Specific to the last sentence in the Abstract: Please see Munchak, et al., 2012: "Relationships between the Raindrop Size Distribution and Properties of the Environment and Clouds Inferred from TRMM", J. Climate, 25, 2963–2978. This is a relevant paper.*

This sentence has been removed from the abstract, as it was not presented in clear language.

*General: Note that MGD is sometimes refers to G-G, see for example, Petty, G. W., and Huang, W. The modified gamma size distribution applied to inhomogeneous and non-spherical particles: Key relationships and conversions. J. Atmos. Sci. 2011: Vol. 68, pp. 1460–1473.*

This is now stated explicitly in the second paragraph of the introduction to be clear on this point.

*Page 2, line 5: Refer to Testud et al here.*

Done

*Page 3, line 31: What is the size resolution of the ODM?*

The size bins used by the ODM are now contained in the text as Table 1. Because there are 120 size bins in the dataset and not all of these are relevant for readers, 60 are given in the table.

*Page 4, line 4: Please clarify "impacts of turbulence"*

To clarify this, the following has been added to S2.1: "A wind vane turns the disdrometer to keep the optical path normal to the wind direction, and the disdrometer's cylindrical volume ensures that the incident angle of hydrometeors does not affect the measurement. These work in concert to minimize impacts of turbulence from local up- and down-drafts, to limit under-catchment and drops impacting the sensor from various directions (Klepp, 2015)."

*Page 4, para 3: It's not clear why the accuracy should depend on oceanic or continental cases. Bumke and Seltman showed that the average DSD shapes were similar in coastal and continental locations. They used the scaling of Sempere-Torres et al to fit the coastal and continental DSDs and found an invariant shape. Perhaps this should also be mentioned here.*

This paragraph was originally added on the suggestion of the editor, to give a sense of the trustworthiness of the derived DSD parameters and whether the accuracy of the derived parameters varies for different regimes. To clarify, we are not suggesting that the accuracy of derived DSD parameters would depend on the location, but rather there may be regime-dependent biases owing to the sensitivity of the instrument that would have an impact on the derived parameters. In the text we have added the following sentence to make this clearer: "In other words, the accuracy DSD parameters reported by OceanRAIN may exhibit bias in regimes with many small drops below the disdrometer's sensitivity threshold or for distributions with a shape unlike that assumed."

*Page 7, end: Recommend plotting Nw vs Dm on semi-log scale to further illustrate the inverse correlation between the two, for ODM and GPM.*

This is now included in the manuscript as the new Fig. 4.

*Page 9, line 5: It's not clear why mu=2 fits the mode of the normalized DSD shape in fig. 3 especially for D/Dm<0.5. In fact mu=-2 appears to be better.*

Interpretation of this figure was a little tricky due to the color scale being saturated and the very high density of points in the middle of the figure. To show this better, the figure's color scale has been modified to hopefully be clearer and more differentiable for readers.

*Page 9, line 15: Please clarify this range for mu for tropical cases. The mu estimate is very sensitive to the shape of the small drop end. It is not clear what the resolution and accuracy are for the ODM at tiny-small sizes. No independent evaluation of the ODM accuracy is given for the small drop end. Hence, caveats are recommended when statements regarding mu ranges are given.*

This has been modified to be less declarative: "...ranging roughly between $\mu = 0$ to $\mu = 3$..." and the point regarding ODM's insensitivity to small drops is well taken. Caveats regarding this insensitivity are now present in various places within the revised manuscript.

*Page 10, line 2: What is meant by "moments" in this context? Are you referring to the moment-based estimation of mu?*

This was unclear and inconsistent in the manuscript, sometimes tending towards the usage in modeling such as single or double moment microphysics. To make this clearer, different versions of the MGD are now specifically defined in Section 2.1 and referred to in terms of 'parameters' used instead of 'moments.'

*Page 11, line 6: use of "power" is not conventional...please use another descriptor.*

This sentence has been modified: "...the MGD with a low $\mu$ value lies near the highest probability densities of the observed PDF."

*Page 11, line 15: the moments fit should be explained earlier.*

This terminology has been modified and is now stated earlier in the manuscript, in Section 2.1.

*Page 11, line 15: Units for RWC are not clear....seems like mg/m^3. No discussion of fig.5 ? Except for second panel, the remaining GMM are not close to the measured N(D).*

The units of the RWC for the figure are now reiterated in the caption to be clearer. Discussion of the original Fig. 5 is now included as a new paragraph, as this was indeed lacking before.

*Page 12, line 2: At this point, recommend that the 4-parameter gamma be used to illustrate that 2 shape parameters are needed.*

The following sentence was added to that paragraph: "It is an indication that a second shape parameter may be useful for describing oceanic DSDs, in line with the generalized gamma approach argued for by Thurai and Bringi (2018)."

*Page 17, line 5: The conclusion is based on the assumption that the resolution and accuracy of N(D) from ODM for small drops is well-established but this has not been demonstrated. The use of total accumulation is not the only criteria by which one justifies the use of the MGD. The width of the mass spectrum is considerably increased when the small drop end is accurate. From cloud physics viewpoint the N(D) is a result of various microphysical processes that are controlled by the low order moments including M0. The GPM algorithms use path integrated attenuation as a constraint...the attenuation is also sensitive to the small drop end especially at Ka-band.*

This is a good point regarding GPM's use of PIA, and this has been added to the description of the measurement vector used by GPM CORRA in Section 2.2.

*Page 17, line 29: What about stratiform vs convective vs shallow rain types?*

This was addressed somewhat in the first paragraph of S3.1, in that from a retrieval perspective, a priori categorization of precipitation into stratiform/convective/other types is highly dependent on ancillary information or just not feasible.

**Reviewer 2**

*1. Reorganize to separate Data Information / Tools (the datasets, their intricacies, and the models you ran them through) from Results (what your methods produced) from Discussion (as a logical conclusion of the results shown, what do the results mean?; what are the implication of results based on the methods you used?). Right now, methods, results, and discussion are mixed throughout each paragraph. This often means the authors often repeat themselves. ... The most egregious instances of mixing discussion with results are in Sec. 3. ...*

In response to this comment, the paper has been restructured to separate results explicitly from discussion. A separate discussion section now exists before the summary and conclusions.

*2. Rewrite/Revise each sentence for literal truth. Each statement should be made to be physically true, word for word. So many of the sentences in this paper contain vague, non-specific, perhaps physically impossible language that is abstract*

*and doesn't really teach the reader anything. The authors need to more specific detail in several places so that their results are digestible, understandable, and reproducible. In many cases, I did not understand what the authors even did in terms of methods, how they drew the conclusions stated, what certain plots really showed, or how to draw conclusions from these plots. In many cases, problems/issues are alluded to (e.g. uncertainty, constraints, sensitivity), but the specifics or causes of these*

5 *problems are withheld so the reader is left with suspense - what could the problems really be and why are they there? We don't know, because the authors allude to issues without explaining them.*

Responses to the reviewer's specific examples, ranging from a-u, follow here: (a) 'easily' has been removed, but the remainder of the sentence stands, as it is a general statement about a statistical method so it does not to be justified in specific physical terms at this stage. (b) It is permissible to use 'underpin' to mean 'support' and the ODM does support the OceanRAIN data

10 collection. (c) The sensitivity of DPR is now given in dBZ in Section 2.2, but the exact sensitivity in terms of drop sizes is not addressed as this is not straightforward. It depends on both size and concentration, and most GPM literature rather quotes minimum detectable rain rates, which is less applicable here for a discussion on DSD. (d) 'more' has been removed. (e) Many instances of 'common' have been replaced with other words. (f) Discontinuity here simply meant bins with zero counts in between bins with positive counts, visualized in Fig. 7. There is no systematic instrument error reported, but we assume that

15 some random error does exist. (g) Removed. (h) The authors feel this is justified despite being metaphorical, because it conveys the sentiment accurately that GMM does not solve the DSD representation issue in one fell swoop. (i) Done. (j) Changed to 'clearly different.' (k) 'Functional form' is used in the manuscript to mean the mathematical construct such as MGD, whereas 'model' and 'fit' are used in different contexts in the manuscript. (l) 'Under/Un-constrained' in retrieval terms means that there is less information than unknowns, now stated explicitly in the text: "...an under-constrained problem (more unknowns than

20 information)..." (m) Changed to "conserve total RWC" (n) Removed 'particularly.' (o) 'This' was changed to "The calculated rain rate" to clarify this sentence. (p) Modified to "higher number concentrations occur over warmer ocean surfaces..." (q) This statement is purposefully vague as it is about various differences, pointed to by the two citations offered and elucidated by the subsequent sentences. (r) All instances of 'bulk' have been removed or modified. (s) Many of these words have been removed or modified in the manuscript. Regarding under- vs. over-representation, it should be clear from the text and figure that it refers

25 to whether the frequency of occurrence is lower or higher than that observed. (t) Posterior is a standard phrase in optimal estimation methodology and needs to be retained. (u) This has been clarified to "not always well represented by the 3-paramter MGD."

*3. I'm unsure about the title after reading the paper. The authors conclude that there are several forms of DSD shape/solution, all of which overlap a bit, and their large datasets include instances of DSD that oscillate around these forms (as shown by*

30 *the machine learning). Then they make conclusions about how well in situ observations of DSDs are represented with current satellite DSD retrievals (sensitivity problems appear to limit the capabilities of the satellites), and how much scatter should exist in satellite retrieved Zh and Tb given the observed DSD (quite a bit, and it varies based on frequency). The paper really doesn't separate out "raindrop regimes"... which would indicate to me to be physical distinctions like monsoon phases, convective v. stratiform, different meteorological conditions, etc. Perhaps the impact and message of the paper is best summarized as: "On*

35 *the distinctiveness of oceanic raindrop distributions in observations and satellite measurements"*

The word 'regime' is indeed quite loaded in this context, and not used specifically enough to be justified. With this in mind, the authors have decided to change the title to "On the distinctiveness of observed oceanic raindrop distributions" and removed the use of 'regime' in that context elsewhere in the manuscript.

*4. In intro: "A lack of globally representative DSD"... do we lack that? What is lacking, specifically? Are you saying that*
5 *we need to find a single DSD that represents all DSD across the globe (implied by the way this is written), or that we need a representative dataset? Unclear. 5. The authors don't give enough credit to the oceanic-ish DSD measurements that are still being reported at Manus Island and Kwajalein Atoll. These are not representative of other DSD in other regions, but they do contribute something important; not all the DSD studies have been confined to land or coast (Thompson et al. 2015 and 2018, and this data is also used in Dolan et al. 2018).*

10 The language here was not clear, and it now reads as simply "global DSD data." In addition, because the issue of a globally representative DSD data set is important to the paper's justification, the following has been added to the end of the paragraph: "It is thus desirable to have measurements of DSDs over ocean, and crucial that these measurements are globally representative rather than skewed toward one region or another." On the second point here the reviewer is quite right, and we now acknowledge the importance of data from these sources: "...observations of DSDs over ocean have mostly been limited to field campaigns, a
15 few small tropical islands and atolls, and coastal radar retrievals."

*6. The authors make up a quantity of RWC and RWP, when it seems that literature to-date uses LWC and LWP to mean the same thing. Paper should conform to these precedents instead of inventing or using their own terminology. It's clear in the paper that only rain cases are used, not ice- or mixed-phase precip, at least at the ground.*

The terminology used is perhaps not standard everywhere, but is quite common for discussing precipitation in retrievals and
20 models, especially when it is desirable to differentiate between hydrometeor types. Indeed the GPM algorithms differentiate between cloud LWP, RWP, GWP, IWP, as these all behave quite differently with regard to the radiative transfer. Our reasoning for this is now given before RWP is defined: "Here we differentiate between cloud water and rainwater due to their different radiative characteristics, with the total liquid water path being the sum of the two."

*7. Fig. 2 - each of the plots must be spaced farther apart in the Latitude dimension so that the distributions can be viewed as*
25 *distinct. The plots run into each other and cannot be distinguished one from the other. Therefore, the point of this plot is lost. Also, It's not clear what the latitude averaging bin width was here. It's centered 20deg apart - is that the center value +/- 10 deg in each direction? Explain in text and caption*

This figure has been changed to a different style of plot that should make interpretation much simpler for readers.

*8. The units of Nw need to be checked - it's often discussed in the text as Nw when it's really logNw that is plotted. The*
30 *parentheses should be adjusted to explain the units properly. log10(Nw [mm-1 mm-3]). Log10Nw doesn't have the same units as Nw itself.*

This has been rectified in the figure caption and the figure itself.

*9. Each of the box and whisker plots need a legend that explain the box width, lines dots, cross hairs (what are the percentiles, etc.). It's too much work to read the caption and try to interpret the plot at the same time.*
35 These are now more explicitly stated in the figure caption.

*10. Add a plot of the map of observations used, so that your Fig. 1 and comparisons between latitude bands can be better understood. The text mentions that more or less data was collected in certain regions, but the reader can't deduce that because no plot of the data extent is shown.*

Done.

*11. Fig. 4 is just about incomprehensible. The conclusions about this figure do not appear to stem logically from the plot, and the plot itself is confusing. If you are making a comparison between models and obs, that is one thing and can be shown a certain way. Then if you also want to compare latitude bands, you should show the PDFs from those latitude bands first before you plot the difference. The difference here doesn't really make sense either because it doesn't seem physical to subtract a tropical PDF from a higher lat PDF; the result doesn't really have any physical significance (or at least the significance is not well-explained or logical). Suggest deleting Fig. 4 and showing the real tropical and high latitude PDFs. Then see if your stated conclusions are supported by the real data. The entire discussion of Fig. 3-4 needs to be carefully curated to be sure that the interpretations of the authors are readily apparent from the plots shown; I couldn't really see how they made all the conclusions that were stated. Particularly the regional dependence, but also the under vs. over "representation" issue. Fig. 4 needs titles to distinguish them and explain more about what is shown.*

This figure has been completely redone to be clearer.

*12. I kept forgetting what GMM and MGD meant. . . these acronyms are used throughout and their physical significance became lost several pages in. Suggest using plain language. See. Fig. 5 - it's not clear what the significance of the acronyms are for the purposes of showing the data.*

While the authors admit that there are many acronyms used in the text, especially around that figure, neither GMM nor MGD can easily be replaced by 'plain language' without something being lost or the text becoming much more wordy. For example, MGD needs to be differentiated from generalized gamma (which is also discussed) and the Petty and Huang (2011) paper we cite uses this acronym. GMM is the standard way to refer to Gaussian Mixture Models in the literature as well, and using plain language such as 'the machine learning technique' would be less specific and more wordy.

*13. Typesetting should be improved a. The lack of line numbers on each line makes it difficult for the reviewer to suggest particular comments and corrections in each case. The authors should number each line in the future. b. The paragraphs would also be easier to distinguish if their indentation was larger. It kind of all runs together c. The references within parentheses are wrong. For example this is typed often: (also used by Duncan et al. (2019)) < see double parentheses at end here. d. Most journals forbid sentences to begin with the name of the variable and will insert words to rectify this situation, whether or not it makes physical sense. The authors are encouraged to fix this the way they want before it gets changed for them (see paragraph above equations on page 2). Completing these sentences also will help ground them in concrete, physical terms: e.g. correcting to be: "The values of N0 and NW. . ."*

The reviewer's first two typesetting suggestions are counter to the style of ACP and the EGU journals. The manuscript was prepared with the ACPD Latex template, and thus the typesetting should be in accordance with ACPD articles. The latter two points, c and d, have been amended in the text wherever found.

*14. Equation (3) is really 3 equations. . . separate them?*

Done

*15. Nw is normalized by LWC. make this clear in its discussion on page 2 and elsewhere.*

Done

*16. Petkovic et al. 2018 reference page 2 - this is not the definitive nor the first paper to study these things. Reference prior work more appropriately to justify this statement - other canonical papers?*

The recent study cited is indeed not canonical and is more relevant for ice hydrometeors anyway. It has thus been removed.

*17. References are made to OceanRAIN-M . . . what is the -M??? Not defined.*

At its first introduction, OceanRAIN-M is now defined: "Specifically, the OceanRAIN-M ("OceanRAIN Microphysics") data..."

*18. The ODM470 doesn't sense drops smaller than 0.4 mm because the voltages are too noisy below this inferred level (the sensitivity is incorrectly stated as 0.3 in the manuscript). From Klepp 2018: "The first 12 bins ranging from 0.04 to 0.36mm in size are not recorded because they are prone to contain artificial signals caused by ship vibration." This is a significant downside to the instrument that needs to be explicitly stated in the manuscript. A large portion of the DSD spectrum is contained below this level (Thompson et al. 2015, Thurai et al. 2018). Also, the ODM assumes a fall velocity and does a vector mean of the fall velocity + wind through sensor. This is an approximation and, in my interpretation of the ODM methods, a potential source of error in the computed number density. 19. "Is an issue faced by all disdrometers" -> yes, but it's the worst for ODM compared to other disdrometers currently used. See comment above.*

The following has been added to address this, as it is indeed worth stressing: "To be clear, there can be significant number concentrations below this sensitivity limit, but voltages corresponding to drops less than $0.36\,\mathrm{mm}$ are disregarded as these can be contaminated by vibrations from the ship (Klepp et al., 2018) and this is a key drawback of the data set."

*20. Minimize effects of turbulence -> minimize under-catchment of drops and minimize the effect of drop splashing against the sensor from multiple directions. Turbulence will still exist. It's just that the orientation of the instrument attempts to catch as many drops as it can.*

This is duly noted and has been corrected. See response to Reviewer 1 above.

*21. Why would the disdrometer NOT "show no difference in accuracy between oceanic and continental cases"? Why would differences occur? It should work the same, except that while at sea, it has to reorient into the wind and a vector mean calculation is done to assume fall velocity+wind through the sensor, which could be susceptible to flow distortion.*

This was also addressed in response to Reviewer 1 above.

*22. Introduction of the "Combined Data" is confusing. The heading of that subsection is also confusing. What does it mean?*

The authors admit that it is confusing, but the GPM project is not consistent when referring to this data set, even internally. The definitive paper by Grecu et al. (2016) is titled "The GPM Combined Algorithm" but never introduces any acronym, whereas some other GPM publications use the acronym CORRA. It is perhaps less confusing if CORRA is introduced as an acronym and used throughout, so that is what we have elected to do. The subsection title has been changed in accordance.

*23. State the different vertical resolutions of the two DPR frequencies. It's mentioned that they differ, but not what they actually are.*

The vertical resolutions do not differ in normal scanning (NS) mode. From Hou et al. (2014): "Both radars have a nominal vertical range resolution of 250 m, sampled every 125 m, with a minimum detectable signal of better than 18 dBZ." In high sensitivity (HS) mode these differ, but we are not using HS data in this study.

*24. The specifications of the GMI and DPR (geometry, resolution) are scattered about the last paragraph of 2.2. Make it more concise.*

These are only scattered this way because they are germane to interpretation of CORRA, whereas earlier description of GPM was about the satellite itself. Readers can consult the cited literature if they desire those details.

*25. "Ground-based data from Ocean-RAIN-M" —— Ocean rain is over ocean! Not land.*

All references to "ground-based" have been changed to "surface-based."

*26. How is clutter classified in satellite retrievals? Sec. 2.2. Seems like it would impact your analysis.*

For the methods of dealing with surface clutter, we will defer to the JAXA documentation on radar processing for DPR. It is not especially important for the discussion here as it is less of a concern over ocean, and DPR's relatively fine 250m vertical resolution means that the data will not witness big vertical jumps from one pixel to another. As we are just using histogram data from the level 3 monthly product and not pixel (level 2) data here, it is deemed of lesser importance for discussion.

*27. Provide citation for GMM upfront at its first introduction*

Done

*28. 90,000 points seems REALLY low compared to the size and length of time of the OceanRAIN dataset. Is this correct? I've collected ODM data over a month and gotten more than 40,000 points of usable data.*

This is correct. From Klepp (2018), 3-4% of the total 6.83 million minutes of observations in OceanRAIN contain rain rates above $0.1\,\mathrm{mm\,h^{-1}}$. In addition, the screening process we use requires that each point be 'rain definite' by the flag provided, contain at least 50 drops measured, and have the MGD fit parameters given. These last two criteria ensure there will be a proper distribution of drops, as OceanRAIN only provides $D_m, N_w, \mu$ if "the PSD has at least 10 size bins filled with data" (Klepp 2018). Thus the total number of cases is only a fraction of the total 'raining minutes' measured by OceanRAIN. To make this clearer, the text has been amended thus: "Only data points marked as rain definite, with 50 or more measured drops, and with a probability of precipitation of 100% were used in the following analysis. To be consistent, only data points with measurements in ten or more size bins are used (Klepp et al., 2018), as these provide the parameters from the MGD fit to Eq. 3."

*29. What are the size ranges of the 60 size bins? And what are their approximate spacing?*

As in response to Reviewer 1 above, a table of the bin sizes is now included as Table 1.

*30. In several places, the authors assert that assumptions and fits haven't been used on DSD obs, but then they say that the data are normalized and that a "nominal shape parameter is assumed" ... This seems contradictory. And the authors fail to mention how the data are normalized - by liquid water content? Similar to Nw? how was the parameter chose?*

Perhaps this is confusing, but it is not contradictory as we are discussing the data used in different contexts. This is now addressed with a separate paragraph in Section 2.1 and clarified in the results section as well when presented.

*31. Provide citation of BIC when introduced*

Done

*32. The authors make claims that they are testing for differences in DSD in different regions or locations, but really they only separated data into very wide latitude bands (Fig. 1, Fig. 2). This distinction, and the implication of the real results, need to be me explicit so as not to overstate the significance or conclusions. Search all uses of "region" and "location" to determine whether you actually mean "latitude" 33. Similarly, the authors make mention of a stratocumulus region, but have not proven where this is or justified their interpretation of that based on a map of the data - which should be added to justify statements made throughout the analysis in Sec. 3*

This is a good point, and all unspecific instances of using 'region' or 'location' have been amended. However, some of the discussion of the distinctiveness of GMM shapes did look into regional occurrences and thus this discussion has been maintained despite not being shown explicitly in this manuscript.

*34. The differences in DSD based on SST seem readily explainable from the Clausius Clapeyron equation - warmer SST leads to higher saturation vapor pressure of air, so moisture content can be higher. – see Fig. 1 and Fig. 2. Nw is directly proportional to LWC (your eq. 3), so result of Fig. 1 lower right is awesome but perhaps not surprising.*

Indeed, and the text has been amended to include mention of C-C: "..., as may be expected due to the Clausius-Clapeyron equation."

*35. "Blue" dots in Fig. 2 don't show up. Looks black.*

Assuming that the reviewer meant Fig. 1, the blue diamonds referenced in the figure caption are outlined in black but filled with blue, which is hopefully clear in the PDF. The updated figure has removed the black outlines.

*36. What are the bin sizes of latitude and SST used to create this plots?*

These are now stated explicitly in the figure caption as 20 degrees latitude and 5 degrees in SST.

*37. Captions and discussions of Fig. 2 say Nw but you plotted log10Nw*

These have been changed wherever appropriate.

*38. The satellite is not sensitive to smallest drops, and probably also not to the lowest number concentrations. This seems to explain some of the differences in flat vs. peaked DSD shape in Fig. 2.*

This is indeed discussed later in the manuscript.

*39. Evaporation below the lowest altitude of GPM would impact the smallest drops first (see Matthew Kumjian et al. papers with observations and models). So, this effect, if it's occurring, would eliminate the small drop portion of the spectrum first, most likely. Since small drops usually present in large number concentrations (Thompson et al. 2015, Dolan et al. 2018), this might mean that the satellite also misses large number concentration DSDs as a result of missing the smallest drops.*

This is true, but the authors suspect that GPM's limited sensitivity to small drops and low number concentrations may be the dominant factor.

*40. The exponential DSD is the most commonly occuring, which is why large combined or averaged datasets of DSD often exhibit a shape of this kind (explained in Bringi and Chandrasekar 2001). However, this book also states why you wouldn't expect high- frequency observations of DSD (such as from 1-min observations) to look exponential. The Marshall Palmer was also based on stratiform, steady, UK rain... so again, it's most representative of this steady-state, averaged, stratiform, weakly-forced rain DSD.*

This is a good point, and factored into our discussion of the histograms from disdrometer observations necessarily looking somewhat different from the large domains observed by GPM.

*41. It is stated on page 10 that rain rate is 3rd moment of DSD; it's actually 3.67th moment. LWC is 3rd moment (Bringi and Chandrasekar 2001).*

The text as written reads: "...with the rain rate calculated as the integral product of the velocity distribution and the third moment of $N(D)$." This is correct as stated. Velocity is proportional to $D^{0.67}$ (Atlas and Ulbrich 1974), which is then integrated in a product with $D^3$ and other factors.

*42. "Result in a small overestimation of rain rates by 0.06 mm/hr or 1.9%" - unclear how this was calculated? At all rain rates? Or overall error? I'm confused how the error in mm/hr can be just one number?*

To see this visually, we can look at a histogram of rain rate differences. This figure was not included as it did not seem a major conclusion of the work, but in the text we just provide the mean difference and point to how for some points, this can be a significant error source in the final calculated rain rate, as originally mentioned in the final section of the manuscript.

*43. Raw observations - it's not really raw. You selected certain sizes, the data have been converted from voltages at fractions of a second to DSD parameters based on A LOT of assumptions contained in the Klepp papers. They are not raw, but they are observations you described already in methods. Just don't use a new qualifier to describe them a different way.*

This is a fair point, and uses of 'raw' have been removed.

*44. How were the "random samples" and "random sampling" and "randomly sampled subsets" performed? Chunks of data taken at random? How much? Explain.*

Every use of the word 'random' in the manuscript means that the Python random function within numpy was used, so cases taken are effectively random. This means in practice that the authors are not cherry-picking any cases or subsets, but rather taking true random samples.

*45. Page 11: "curve was chosen"... I just see lots of curves on Fig. 5. How was it chosen?*

This has been amended to be more specific: "...chosen, judged by the highest posterior probability."

*46. Last sentence of 4.1 - was this also seen by Dolan et al. 2018?*

Indeed it was. We have added the following to conclude that sentence: "... and consistent with findings from Dolan et al. (2018)."

*47. Text at top of Page 12: it seems that you are trying to say that all GMM output DSD fits depart from the exponential form, so that observed mu almost never equals zero.*

Yes, or at least that a pure exponential is not a dominant mode of the distributions observed. To emphasize this point, the following sentence has been added: "Indeed, the distributions produced by GMM seldom resemble a pure exponential DSD."

*48. FIg. 6 - label legend better in terms of GMM and MGD parameters (and explain in plain language, it's not clear here). In caption, differentiate solid and dashed lines. Water content should be LWC. Keep consistent terminology/abbreviation throughout.*

The caption has been modified to state which lines correspond to GMM and MGD, and 'water content' has been changed to 'RWC.'

*49. Page 13: rare to have -> rare to observe; observations -> minutes*

Done

*50. Last few sentences before Fig. 7 does not make any sense; overly literary and wordy. And it's not clear how these conclusions can be drawn from what is shown.*

This section has been trimmed and clarified: "In other words, DSDs featuring a strong peak near $D_m$, and for which an exponential is a very poor approximation, are not very common. This can also be seen in Fig. 3, as the PDF is relatively weak in the bottom left of that plot."

*51. It's not clear how interpretation of BIC in Fig. 7 leads to the conclusion made by the authors. I'm totally confused how there can be a singular set... seems like an oxymoron and I don't know what this is intended to mean. It actually appears like 2 modes describe most of the variability (highest BIC) and then Ngmm from 3-6 does fairly and equally well.*

To be clear, the lowest BIC is desirable, as stated in S2.3: "The minimum BIC thus signifies the optimal K value, maximizing the variability explained with the fewest possible classes." And yes, a singular set (or an optimal set) may well be an impossible outcome to expect but that was something we wanted to investigate here, with the conclusion that such a set does not appear to exist or is at least not very clear.

*52. The last paragraph of Sec 4 is very problematic and hard to follow, and hard to deduce the conclusion from the results shown. The claims appear unsubstantiated as written. I don't know what "decoupled from RWC" . . . there was no regional comparison performed. . . use of "not particularly" and " not especially distinct" are handwavy and should be replaced by quantifiable and specific terms.*

This paragraph has been reworked, and the use of adverb qualifiers reduced to clarify these statements. The difficulty in conveying these ideas is that it is inherently not quantifiable that a singular set of DSD shapes does not exist, or that one region or latitude band is not best described by a particular shape, as these are subjective conclusions (though for the latter the plot of BIC provides some evidence). Instead we endeavor to be as specific as possible without overstating our conclusions. With fuller sampling from OceanRAIN these conclusions could be stronger.

*53. What are "variational" systems? Page 14*

'Variational' refers to both retrievals and data assimilation systems that minimize a cost function in an iterative manner, e.g. 1DVAR, 3DVAR, 4DVAR. To make this clear for readers the text has been amended: "...biases in variational systems (e.g. 1DVAR, 3DVAR) if not..."

*54. "Water content" is stated. . . but else where its RWC. stick with consistent terminology/abbreviation once defined, otherwise it sounds like you are introducing a new concept or field.*

This has been changed to RWC in all instances.

*55. Page 14: "was used to perform forward model simulations" ... of what? Incomplete sentence.*

It is not an incomplete sentence, and 'forward model simulations' is a common way to refer to radiative transfer. But it has been rewritten: "Forward model simulations of the radiative transfer were performed using the Atmospheric Radiative Transfer Simulator (ARTS) version 2.3 (Eriksson et al., 2011; Buehler et al., 2018)."

*56. Page 14: what are the custom distributions, or habits? Mentioned but not explained sufficiently.*

Amended thus, removing references to habits since we do not use mixed-phase or ice hydrometeors here: "The ARTS model can handle custom particle size distributions (such as observational size bin data) as well as prescribed DSDs such as the MGD."

*57. Unclear what g m-3 cloud looks like or what this number is supposed to represent.*

This is a cloud liquid water path value, quite common in the remote sensing literature as exemplified by the reference given to Lebsock et al. (2008). Because it is only mentioned once, the authors thought it best to not introduce another acronym (CLWP) and instead give the column mass and depth of the cloud layer. This was similarly done in Duncan et al. (2018).

*58. Define TESSEM2*

Done

*59. How was the change in TB computed? From level to level or from certain simulations? The description of Fig. 8 and these simulations is very confusing. I don't really know what you are trying to do or how you are doing it. Clarify. What are the units of TB here? What about units of RWP? Units need to be stated in the text. I don't understand what impacts are being tested here , "The 89 GHz shows little impact" . . . impact of what on what?*

This was computed as the top of atmosphere change relative to RWP of 0, now stated explicitly in the text. TB is given in Kelvin, and its change in $\Delta K$, as is standard. "Impact" has been changed to "sensitivity" to be more specific. RWP is given in units of $\mathrm{kg\,m^{-1}}$, now stated explicitly when first introduced.

*60. What do you mean by"largely cancels out its emission signal?" Page 14*

Radiative transfer impacts are usually referred to in terms of emission signals and scattering signals, in this case positive and negative impacts on measured $T_B$. This statement means that the impact on $T_B$ from scattering is about equal to the impact from increased emission, and so these cancel out. This sentence was slightly modified for clarity: "...its signal is mainly from cloud water emission, and the scattering signal from rain largely cancels out its emission signal from rain."

*61. What are the sizes of cloud droplets and rain droplets that produce these stated differences in emission vs. scattering?*

This information has been added: "Cloud droplets are monodisperse with diameter $15\,\mu\mathrm{m}$, whereas the rain drops are about two orders of magnitude larger in diameter, hence the differing scattering properties."

*62. Fig 8 and Fig. 9 and other figures where error bars (of some kind?) are shown - need to define on legend what the errors bars and whiskers and dots really mean.*

It is already stated in the caption of both figures that means and standard deviations are shown. We now state explicitly in the caption that the means are given by dots and the standard deviations by bars.

*63. Add titles to Fig 8 so that you can tell from looking at the plot what the differences between a and b are.*

Done

*64. Fig. 8 discussion: "net response" should be "mean value" according to your plot. Be specific about what you are discussing. I'm confused what "response" means in this context.*

Here 'net response' has been changed to 'mean value.'

*65. The averaging window of 6 min seems too short to prove your point. Assume that ships transit at 10 kt (nautical miles per hour) - a good assumption, or you can test it with all the meta data in the ODM dataset. Anyway, figure out how long in*

*time you need to average in order to approximate a 5 km or degree or whatever distance spanned by a single pixel of satellite data*

So, 10kt is 18.52kmph, so it would take about 16min to traverse a 5km distance at that speed. The data have been reanalyzed using this time window, and the figure updated: "Specifically, a nominal 16 minute window was used to average consecutive raining disdrometer measurements, in that a ship at $10\,\mathrm{kn}$ would take about 16 minutes to traverse $5\,\mathrm{km}$."

*66. "Should not be markedly different." . . .. If what? Compared to what? A comparison or warning seems to be made, but it's not clear what it is.*

The language here has been amended: "The maximum forward model errors observed by a sensor such as GMI may not be markedly different than those presented with the time averaging performed, however most GMI channel footprints are larger than that of DPR."

*67. "Non-raining points" – do you mean that you did not average zero values? Unclear*

This has been clarified: "Observations with zero rain rates.."

*68. Fig. 9 - why aren't data shown above 40 mm/h? Surely they exist in ODM?*

This figure has been extended up to now include the largest bin at $46\,\mathrm{mm\,h^{-1}}$. The data become noisy at the higher rain rates due to a low sample size (the final bin shown contains only 17 observations), and this is now noted in the text as well.

*69. The author's conclusion that the error is way bigger for the blue line is not well- supported. The errors are only different in blue v. red at the very highest rain rates (above 10 mm/hr). Consider revising interpretations for clarity / consistency.*

In fact, the standard deviations shown at $K_A$ are smaller at every single rain rate shown. They may not be much different at the lower end, but they are still smaller.

*70. Page 16 "largely been predicated on limited land-based observations in the past" - including some island observations from Manus, Gan, and Kwajalein.*

The key word here is 'largely' and this seems to be true.

*71. Page 16: here it is stated that DSD have limited dependence exists on SST, but in presentation of Fig. 1 you remarked that there appeared to be an SST dependence. Inconsistent.*

This is consistent, because despite there being some dependence, it is indeed quite limited.

*72. Page 16 in several places: datasets "observe" -> exhibit. Not literally true.*

Changed

*73. Summary and Conclusions: your latitude bands are very wide and may inhibit your ability to draw conclusions about DSD variability that could be present in regimes such as the monsoon, ITCZ and SPCZ, subtropical highs, western boundary currents that are really warm, cold current regions, etc.*

To check this, this figure was remade with tighter latitude bands, seen below. The conclusions do not really change, but some bands are much more sparsely populated:

*74. "Function to encapsulate" and "appears more applicable" – > unclear. Revise.*

[Figure]

**Figure 1.** As in Fig. 1 of the original manuscript (now Fig. 2), but with tighter latitude and SST bins.

Revised thus: "Usage of the normalized gamma function to describe all observed DSD behavior was questioned, as it appears more applicable in the Tropics than for higher latitude populations, as high latitude cases exhibit larger concentrations of small drops that are outside the state space specified by the 3-parameter MGD (Fig. 11)."

*75. Line 3 page 17: spatial and temporal considerations*

5      Done

*76. Unclear what this sentence means "Its use can cause systematic. . ..*

This has been revised: "The 3-parameter MGD can cause systematic biases in rain rate estimation relative to using the observed size bin data, quantified to be a -2% error in the mean relative to total accumulation calculated from the disdrometers."

*77. Sentence with "for about 3% of cases . . ." = run-on sentence. Revise to simplify it.*

10     The comma has been replaced by a semicolon.

*78. Page 17 lines 12-24 make very little sense. Revise for clarity a. Uncertainty = standard deviation? Width of distribution? means? b. What is "radiative uncertainty"? c. How "rapidly" d. What is meant by low vs. high frequencies? e. "Half the*

*radiative signal" - do you mean the mean? f. What is "true for passive and active simulations" ?? uncler based on what has been shown or discussed g. Unclear why "retrieving the DSD would shrink these ranges" unsupported from what is shown or explained, or maybe it just doesn't make sense h. Lines 19-24 don't really make any sense. Revise. Unclear. i. It's confusing that you bring up dewpoint temp and RWC on line 28 because you didn't discuss Td anwhwere else but you did use RWC.*

5  *j. What is meant by "limited spatiotemporal sampling of OceanRAIN"? – Please include a map in the revised manuscript k. What references can you provide to justify why DSDs are more uniform over land. They seem to exhibit more modes over land because hail and mixed-phase and very intense convective microphysical processes can occur. Unclear from what is written.*

This section has been significantly modified to use more exact language.

**Correspondence:** David Ian Duncan (david.duncan@chalmers.se)

**Abstract.** Representation of the drop size distribution (DSD) of rainfall is a key element of characterizing precipitation in models and retrievals, with a functional form necessary to calculate the precipitation flux and the drops' interaction with radiation. With newly available oceanic disdrometer measurements, this study investigates the validity of commonly used DSDs, potentially useful a priori constraints for retrievals, and the forward model errors caused by impacts of DSD variability. These data are also compared to with leading satellite-based estimates of oceanic DSDs, indicating that the disdrometers observe more small drops and more variable number concentrations. 
[revised manuscript text omitted]

[Figure]

**Figure 5.** As in Fig. 3, but differences Normalized histograms of PDFs. The left panel $D_m$ (aleft) shows the MGD-fitted PDF subtracted from the full OceanRAIN PDF shown in Fig. 3. The right panel and $log_{10}(N_w)$ (bright) shows the difference of OceanRAIN PDFs from high latitude for GPM CORRA and tropical oceanic locationsOceanRAIN, vizseparated by latitude. $PDF_{>50°} - PDF_{20°N-20°S}$, given as All histograms use a percent differencelinear y-axis of height 20%. Areas in gray indicate no GPM data in one or both PDFs. The low and high $\mu$ curves given approximately bound are from the PDF space for the MGD-fitted data3B CMB monthly gridded product.

[revised manuscript text omitted]